# Benchmarking challenging small variants with linked and long reads

## Graphical abstract

## Authors

Justin Wagner, Nathan D. Olson, Lindsay Harris, ..., Andrew Carroll, Marc Salit, Justin M. Zook

## Correspondence

justin.wagner@nist.gov (J.W.), justin.zook@nist.gov (J.M.Z.)

## In brief

Analogous to placing puzzle pieces that look similar, mapping sequences to regions of the genome that look similar is challenging. Wagner et al. describe a new Genome in a Bottle Consortium resource for benchmarking accuracy of human genome sequencing in more challenging regions of the genome.

## Highlights

- The Genome in a Bottle Consortium presents an expanded benchmark for 7 genomes

- Long and linked reads expanded the benchmark to regions challenging for short reads

- The expanded regions include challenging medically relevant genes like *PMS2*

- This enables development of new technologies and bioinformatics methods

Wagner et al., 2022, Cell Genomics 2, 100128
May 11, 2022

# Cell Genomics

## Article

# Benchmarking challenging small variants with linked and long reads

Justin Wagner,[1,*] Nathan D. Olson,[1] Lindsay Harris,[1] Ziad Khan,[2] Jesse Farek,[2] Medhat Mahmoud,[2] Ana Stankovic,[3] Vladimir Kovacevic,[3] Byunggil Yoo,[4] Neil Miller,[4] Jeffrey A. Rosenfeld,[5] Bohan Ni,[6] Samantha Zarate,[6] Melanie Kirsche,[6] Sergey Aganezov,[6] Michael C. Schatz,[6] Giuseppe Narzisi,[7] Marta Byrska-Bishop,[7] Wayne Clarke,[7] Uday S. Evani,[7] Charles Markello,[8] Kishwar Shafin,[8] Xin Zhou,[9] Arend Sidow,[10,11] Vikas Bansal,[12] Peter Ebert,[13] Tobias Marschall,[13] Peter Lansdorp,[13] Vincent Hanlon,[14] Carl-Adam Mattsson,[14] Alvaro Martinez Barrio,[15] Ian T. Fiddes,[15] Chunlin Xiao,[16] Arkarachai Fungtammasan,[17] Chen-Shan Chin,[17] Aaron M. Wenger,[18] William J. Rowell,[18] Fritz J. Sedlazeck,[2] Andrew Carroll,[19] Marc Salit,[20,21] and Justin M. Zook[1,21,22,*]

[1]Material Measurement Laboratory, National Institute of Standards and Technology, 100 Bureau Dr, MS8312, Gaithersburg, MD 20899, USA
[2]Human Genome Sequencing Center, Baylor College of Medicine, One Baylor Plaza, Houston, TX 77030, USA
[3]Seven Bridges, Omladinskih brigada 90g, 11070 Belgrade, Republic of Serbia
[4]Children's Mercy Kansas City, Kansas City, MO, USA
[5]Rutgers Cancer Institute of New Jersey, New Brunswick, NJ, USA
[6]Department of Computer Science, Johns Hopkins University, Baltimore, MD, USA
[7]New York Genome Center, 101 Avenue of the Americas, New York, NY, USA
[8]University of California at Santa Cruz Genomics Institute, 1156 High Street, Santa Cruz, CA, USA
[9]Department of Computer Science, Stanford University, Stanford, CA 94305, USA
[10]Department of Pathology, Stanford University, Stanford, CA 94305, USA
[11]Department of Genetics, Stanford University, Stanford, CA 94305, USA
[12]Department of Pediatrics, University of California, San Diego, La Jolla, CA 92093, USA
[13]Institute of Medical Biometry and Bioinformatics, Medical Faculty, Heinrich Heine University Düsseldorf, 40225 Düsseldorf, Germany
[14]Terry Fox Laboratory, BC Cancer Research Institute and Department of Medical Genetics, University of British Columbia, Vancouver, BC, Canada
[15]10X Genomics, Pleasanton, CA 94588, USA
[16]National Center for Biotechnology Information, National Library of Medicine, National Institutes of Health, 8600 Rockville Pike, Bethesda, MD 20894, USA
[17]DNAnexus, Inc., Mountain View, CA 94040, USA
[18]Pacific Biosciences, Menlo Park, CA 94025, USA
[19]Google Inc., 1600 Amphitheatre Pkwy., Mountain View, CA 94040, USA
[20]Joint Initiative for Metrology in Biology, SLAC National Laboratory, Stanford, CA, USA
[21]Senior author
[22]Lead contact
*Correspondence: justin.wagner@nist.gov (J.W.), justin.zook@nist.gov (J.M.Z.)

## SUMMARY

Genome in a Bottle benchmarks are widely used to help validate clinical sequencing pipelines and develop variant calling and sequencing methods. Here we use accurate linked and long reads to expand benchmarks in 7 samples to include difficult-to-map regions and segmental duplications that are challenging for short reads. These benchmarks add more than 300,000 SNVs and 50,000 insertions or deletions (indels) and include 16% more exonic variants, many in challenging, clinically relevant genes not covered previously, such as *PMS2*. For HG002, we include 92% of the autosomal GRCh38 assembly while excluding regions problematic for benchmarking small variants, such as copy number variants, that should not have been in the previous version, which included 85% of GRCh38. It identifies eight times more false negatives in a short read variant call set relative to our previous benchmark. We demonstrate that this benchmark reliably identifies false positives and false negatives across technologies, enabling ongoing methods development.

## INTRODUCTION

Advances in genome sequencing technologies have continually transformed biological research and clinical diagnostics, and benchmarks have been critical to ensure the quality of the sequencing results. The Genome in a Bottle Consortium (GIAB)

developed extensive data[1] and widely used benchmark sets to assess the accuracy of variant calls resulting from human genome sequencing.[2–4] To use these benchmarks, the Global Alliance for Genomics and Health (GA4GH) Benchmarking Team developed tools and best practices for benchmarking.[5] These benchmarks and benchmarking tools helped enable the

**Table 1. Summary comparison of v.3.3.2 and v.4.2.1 HG002 benchmark sets**

| Reference Build | Benchmark set | Reference included | SNVs | Indels | Base pairs in seg dups and low mappability |
|---|---|---|---|---|---|
| GRCh37 | v.3.3.2 | 87.8% | 3,048,869 | 464,463 | 57,277,670 |
| GRCh37 | v.4.2.1 | 94.1% | 3,353,881 | 522,388 | 133,848,288 |
| GRCh38 | v.3.3.2 | 85.4% | 3,030,495 | 475,332 | 65,714,199 |
| GRCh38 | v.4.2.1 | 92.2% | 3,367,208 | 525,545 | 145,585,710 |

Shown are metrics calculated for chromosomes 1–22 in GRCh37 and GRCh38, with variant counts and inclusion of segmental duplications (seg dups) and regions that appear similar to short reads (i.e., "low mappability" regions where 100-bp read pairs have 2 or less mismatches and a 1 or less indel difference from another region of the genome).

development and optimization of technologies and bioinformatics approaches, including linked reads,[6] highly accurate long reads,[7] deep-learning-based variant callers,[8,9] graph-based variant callers,[10] and *de novo* assembly.[11,12] As these new technologies and methods accessed increasingly challenging regions of the genome, studies highlighted many known medically relevant genes that were excluded from these previous benchmarks.[7,13–15] These studies demonstrated the need for improved benchmarks covering segmental duplications, the major histocompatibility complex (MHC), and other challenging regions. A separate synthetic diploid benchmark was generated from assemblies of error-prone long reads for two haploid hydatidiform mole cell lines, but this had limitations in terms of cell line availability and small insertion or deletion (indel) errors because of the high error rate of the long reads.[16]

Many of the difficult regions of the genome lie in segmental duplications and other repetitive elements. Linked reads have been shown to have the potential to expand the GIAB benchmark by 68.9 Mbp to some of these segmental duplications.[6] A circular consensus sequencing (CCS) method was recently developed that enables highly accurate 10- to 20-kb-long reads.[7] This technology identified a few thousand likely errors in the GIAB benchmark, mostly in long interspersed nuclear elements (LINEs). It also had more than 400,000 variants in regions mappable with long reads but outside the benchmark, and it covered many difficult-to-map, medically relevant genes that are challenging to call using short reads or lower-accuracy long reads. GIAB recently used these data to produce a local diploid assembly-based benchmark for the highly polymorphic MHC region.

Here we use linked reads and long reads to expand GIAB's benchmark to include challenging genomic regions for the GIAB pilot genome NA12878 and the GIAB Ashkenazi and Han Chinese trios from the Personal Genome Project, which are more broadly consented for genome sequencing and commercial redistribution of reference samples.[17] We more carefully exclude segmental duplications that are copy number variant (CNV) in the GIAB samples[18] or missing copies in GRCh37 or GRCh38[19,20] because these currently cannot be reliably benchmarked for small variants. We also refined the methods used to produce the diploid assembly-based MHC benchmark[21] to include most of the MHC region in each member of the trio. We show that our benchmark reliably identifies false positives (FPs) and false negatives (FNs) across a variety of short-, linked-, and long-read technologies. The benchmark has already been used to develop and demonstrate new variant callers in the precisionFDA Truth Challenge V2.[22]

## RESULTS

### The new benchmark covers more of the reference, including many segmental duplications

GIAB previously developed an integration approach to combine results from different sequencing technologies and analysis methods, using expert-driven heuristics and features of the mapped sequencing reads to determine at which genomic positions each method should be trusted. This integration approach excludes regions where all methods may have systematic errors or locations where methods produce different variants or genotypes and have no evidence of bias or error. The previous version (v.3.3.2) primarily used a variety of short-read sequencing technologies and excluded most segmental duplications.[4] Our new HG002 v.4.2.1 benchmark adds long and linked reads to cover 6% more of the autosomal assembled bases for GRCh37 and GRCh38 than v.3.3.2 (Table 1). The median coverage by linked- and long-read datasets for each genome is shown in Table S1. We also replace the mapping-based benchmark with assembly-based benchmark variants and regions in the MHC.[21] v.4.2.1 adds more than 300,000 single nucleotide variants (SNVs) and 50,000 indels compared with v.3.3.2. In STAR Methods, we detail the creation of the v.4.2.1 benchmark, including using the new long- and linked-read sequencing data in the GIAB small variant integration pipeline and identifying regions that are difficult to benchmark, including potential large duplications in HG002 relative to the reference as well as problematic regions of GRCh37 or GRCh38.

Many of the benchmark regions expanded by v.4.2.1 are in segmental duplications and other regions with low mappability for short reads (Figures 1 and S1; Table 1). GRCh38 has 270,860,615 bases in segmental duplications and low-mappability regions (regions difficult to map with paired 100-bp reads) on chromosomes 1–22, including modeled centromeres. v.4.2.1 covers 145,585,710 (53.7%) of those bases, whereas v.3.3.2 covers 65,714,199 bases (24.3%). However, v.4.2.1 still excludes some difficult regions and structural variants (SVs); of the bases in GRCh38 chromosomes 1–22 not covered by v.4.2.1, segmental duplication and low-mappability regions account for 56.4% of those bases.

To identify the types of genomic regions where v.4.2.1 gains and loses benchmark variants relative to v.3.3.2, we compared the variant calls in v.4.2.1 with v.3.3.2 and used the v.2.0 GA4GH/GIAB stratification files.[22] Figure 1B highlights stratified genomic regions with the largest SNV gains and losses in v.4.2.1

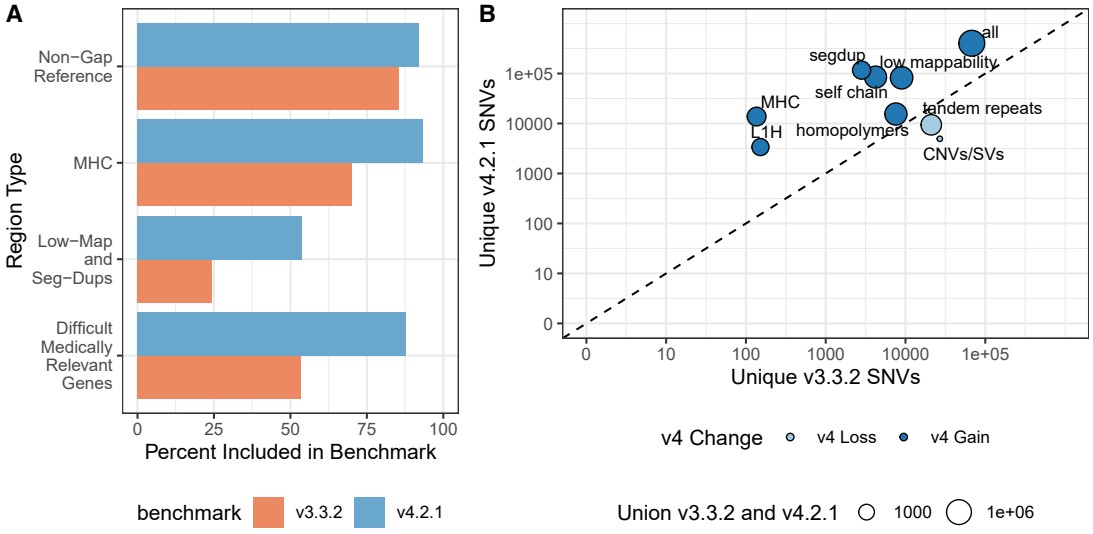

**Figure 1. The new benchmark set includes more of the reference genome and more variants**

(A) Percentage of the genomic region that is included by HG002 v.3.3.2 and v.4.2.1 of all non-gap, autosomal GRCh38 bases; the MHC; low-mappability regions and segmental duplications; and 159 difficult-to-map, medically relevant genes described previously.

(B) The number of unique SNVs by genomic context. Circle size indicates the total number of SNVs in the union of v.3.3.2 and v.4.2.1. Circles above the diagonal indicate a net gain of SNVs in the newer benchmark, and circles below the diagonal indicate a net loss of SNVs in the newer benchmark.

versus v.3.3.2 (the full information is available in Table S2). As expected, the inclusion of linked and long reads leads to more variants in v.4.2.1 than v.3.3.2 in segmental duplications, self-chains, the MHC region, as well as other regions that are difficult to map with short reads. The gain in v.4.2.1 relative to v.3.3.2 is lower in tandem repeats and homopolymers because v.4.2.1 excludes any tandem repeats and homopolymers not completely included by the benchmark regions. Partially included tandem repeats and homopolymers in v.3.3.2 caused some errors in benchmarking results when v.3.3.2 missed variants in the repeat but outside the benchmark regions, so partially included repeats were completely excluded in v.4.2.1.

In addition to including more difficult regions, v.4.2.1 corrects or excludes errors in v.3.3.2. In previous work, variants called from PacBio HiFi were benchmarked against v.3.3.2, and 60 SNV and indel putative false positives were manually curated, which identified 20 likely errors in v.3.3.2.[7] All 20 errors were corrected in the v.4.2.1 benchmark or removed from the v.4.2.1 benchmark regions. Twelve of these errors in v.3.3.2 result from short reads that were only from one haplotype because reads from the other haplotype were not mapped because of a cluster of variants in a LINE; two of these v.3.3.2 errors are excluded in v.4.2.1, and 10 variants are correctly called in v.4.2.1 (Table S3). To verify the v.4.2.1 variants incorrectly called by v.3.3.2 in LINEs, we confirmed all 274 tested variants in 4 LINEs across the 7 samples using long-range PCR followed by Sanger sequencing, as described in STAR Methods and Table S4.

### The new benchmark includes additional challenging genes

To focus analysis on potential genes of interest, we analyzed inclusion of genes previously identified to have at least one exon

that is difficult to map with short reads, which we call "difficult-to-map, medically relevant genes."[13] v.4.2.1 covers 88% of the 10,009,480 bp in difficult-to-map, medically relevant genes on primary assembly chromosomes 1–22 in GRCh38, much larger than the 54% covered by v.3.3.2 (Table S5). 3,913,104 bp of the difficult-to-map, medically relevant genes lie in segmental duplication or low-mappability regions. The v.4.2.1 benchmark includes 2,928,012 bp (74.8%) of those segmental duplications and low-mappability regions, whereas the v.3.3.2 benchmark includes 208,882 bp (5.3%). Future work will be needed to include 49 of the 159 genes on chromosomes 1–22 that still have less than 90% of the gene body included (Figure 2A), such as a recently published assembly-based approach.[20] For example, 5 genes that have potential duplications in HG002 were previously partially included in v.3.3.2 but are excluded in v.4.2.1 because new methods will be needed to resolve and represent benchmark variants in duplicated regions (Figure 2B). The medically relevant gene *KIR2DL1* was partially included in v.3.3.2 but is now completely excluded because the copy number variable KIR region is removed from the v.4.2.1 benchmark regions. v.4.2.1 also does a better job excluding regions that are duplicated in the benchmark sample relative to the reference, specifically because it excludes regions with higher-than-normal PacBio HiFi and/or Oxford Nanopore (ONT) coverage (Figure 3). We detail the inclusion of each difficult-to-map, medically relevant gene in Table S6.

*PMS2* is an example of a medically important gene involved with DNA mismatch repair that is included more by v.4.2.1 (85.6%) than by v.3.3.2 (25.9%) for HG002 (Figure 4). Variant calling in *PMS2* is complicated by the presence of the pseudo-gene *PMS2CL*, which contains identical sequences in many of the exons of *PMS2* and is within a segmental duplication.[23] Using long-range PCR followed by Sanger sequencing, we

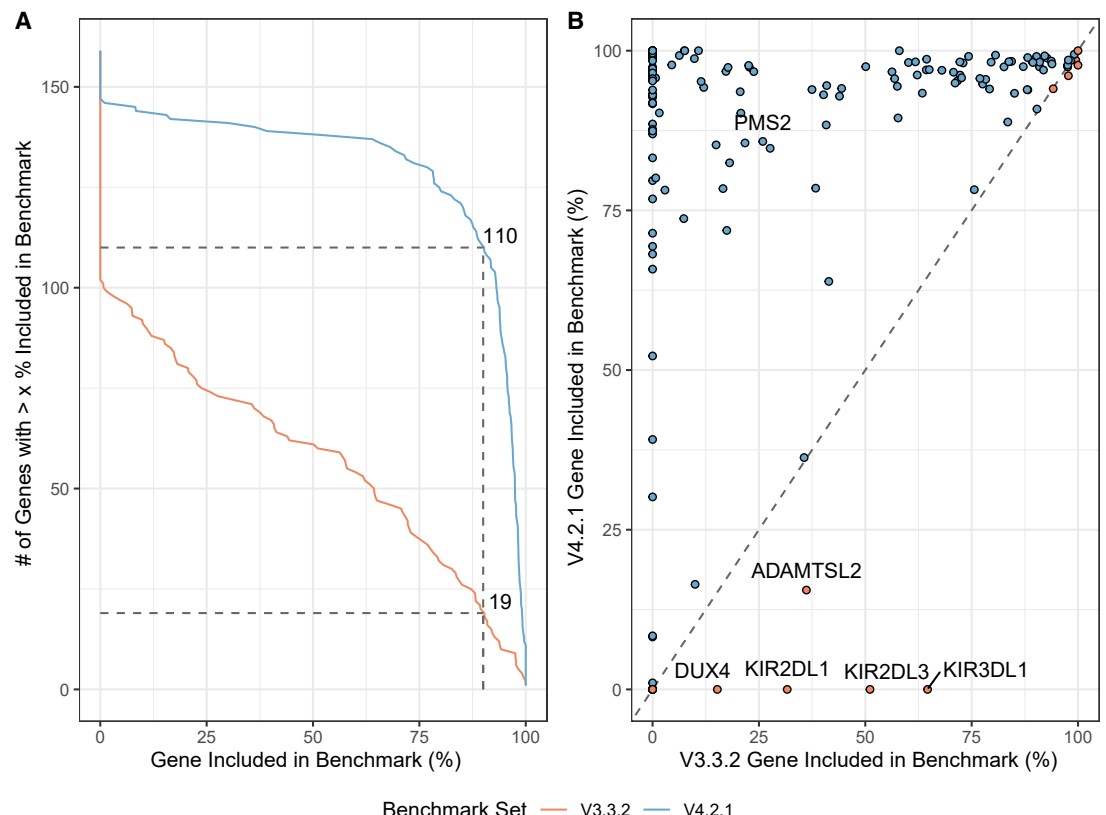

**Figure 2. v.4.2.1 includes many more difficult-to-map, medically relevant genes**

(A) Cumulative distribution of the percentage of each gene included in HG002 v.4.2.1 benchmark regions for 159 autosomal difficult-to-map, medically relevant genes. Dashed lines indicate that the number of genes included more than 90% increased from 19 in v.3.3.2 to 110 in v.4.2.1.

(B) Pairwise comparison of difficult-to-map, medically relevant gene inclusion in the benchmark set. Genes falling on the dashed line are similarly included by both benchmark sets, whereas genes above (red fill) or below (blue fill) the dashed line are included more by the v.4.2.1 or v.3.3.2 benchmark set, respectively. The genes included more by v.4.2.1 tend to be in segmental duplications, and the smaller number of genes included more by v.3.3.2 are mostly genes duplicated in HG002 relative to GRCh38 and should be excluded.

confirmed 1,516 v.4.2.1 benchmark variants in *PMS2* and 20 other difficult-to-map, medically relevant genes across the 7 samples, and only 4 in *PKD1* and 1 in *FCGR2B* were discordant with Sanger. The 5 discordant variants appeared to be clearly supported by short and long reads, and the reason for the discordant Sanger result was unclear. Detailed Sanger sequencing results for each gene and sample are shown in Table S4.

## Comparison with platinum genomes identifies fewer potential errors in v.4.2.1

Platinum Genomes identified SNVs that were Mendelian inconsistent because of being called heterozygous in all 17 individuals in a pedigree with short read sequencing ("category 1" errors).[24] Some of these heterozygous calls result from regions duplicated in all individuals in the pedigree relative to GRCh37. Therefore, category 1 SNVs matching SNVs in our benchmarks may identify questionable regions that should have been excluded from the benchmark regions. 326 category 1 SNVs matched HG002 v.4.2.1 SNVs, a decrease relative to the 719 category 1 SNVs matching HG002 v.3.3.2 SNVs. This suggests that v.4.2.1 better excludes duplications in HG002 relative to the reference even as

it expands into more challenging segmental duplication regions. However, the remaining 326 matching SNVs may be areas for future improvement in v.4.2.1. Manual curation of 10 random SNVs in HG002 v.4.2.1 that matched category 1 variants showed that 5 were in possible duplications that potentially should be excluded, and 5 were in segmental duplication regions that may have been short read mapping errors or more complex variation in segmental duplications (Table S7). Particularly, clusters of v.4.2.1 variants matching category 1 variants appeared to be likely errors in v.4.2.1. We also compared the v.4.2.1 HG001 benchmark with the 2017 hybrid short-read benchmark from Platinum Genomes, which uses an orthogonal approach based on including variants with genotypes phased as expected in the 17-member pedigree. The concordance between v.4.2.1 and Platinum Genomes in the intersection of both benchmark regions was 99.96% on GRCh37 and GRCh38. Curation identified most differences as likely short-read mapping biases in Platinum Genomes because 454 of 654 GIAB-specific and 1,857 of 2,203 Platinum Genomes-specific variants on GRCh37 fell in low-mappability regions and segmental duplications. In addition, relative to the short-read-based Platinum Genomes benchmark

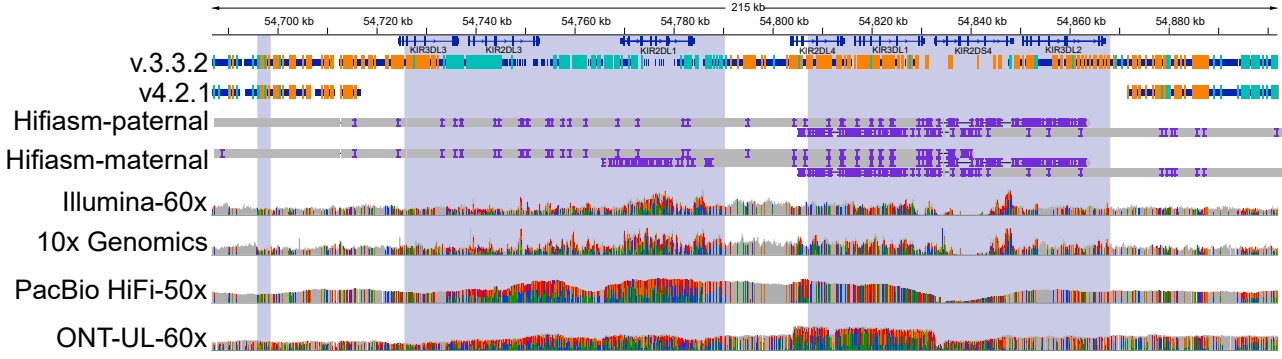

**Figure 3. Genes in the KIR locus are excluded in v.4.2.1 because of duplication in HG002**

Medically relevant genes in the KIR locus, such as *KIR2DL1*, were partially included in v.3.3.2 with many erroneous variants but are correctly excluded by v.4.2.1 because of a likely duplication and other structural variation. Thick blue bars indicate regions included by each benchmark, and orange and light blue lines indicate positions of homozygous and heterozygous benchmark variants, respectively. A duplication of part of this region, which is common in the population, is supported by higher-than-normal coverage and high variant density across all technologies as well as alignment of multiple contigs from the maternal trio-based HG002 Hifiasm assembly (Hifiasm-maternal). The region is very challenging to characterize and assemble accurately because of high variability and copy number polymorphisms in the population as well as segmental duplications (shaded regions).

regions, the v.4.2.1 benchmark regions have substantially fewer small gaps that can cause problems when benchmarking[4] so that the NG50 size of benchmark regions in v.4.2.1 is more than two times greater than Platinum Genomes (Figure S2).

## High Mendelian consistency in trios

To further evaluate the accuracy of the benchmark, we evaluated the Mendelian consistency of our v.4.2.1 benchmark sets for the son, father, and mother in two trios from GIAB of Ashkenazi ancestry (HG002, HG003, and HG004) and Han Chinese ancestry (HG005, HG006, and HG007). In the intersection of the benchmark regions for the Ashkenazi trio, this evaluation identified 2,502 variants that had a genotype pattern inconsistent with Mendelian inheritance of the 4,968,730 variants in at least one member of the trio (0.05%), slightly below the rate for v.3.3.2 (2,494 of 4,383,371 or 0.06%) on GRCh38. The Mendelian inconsistency rates for the GIAB Han Chinese trio were lower than for the Ashkenazi trio, 821 of 4,601,643 (0.02%) for v.4.2.1 and 790 of 4,138,328 (0.02%) for v.3.3.2. We separately analyzed Mendelian inconsistent variants that were potential cell line or germline *de novo* mutations (that is, the son was heterozygous and both parents were homozygous reference) and those that had any other Mendelian inconsistent pattern (which are unlikely to have a biological origin). Of 2,502 violations in HG002, 1,177 SNVs and 284 indels were potential *de novo* mutations, 67 more SNVs and 71 more indels than in v.3.3.2.[4] HG005 had only 162 potential *de novo* SNVs and indels. Following manual inspection of 10 random *de novo* SNVs in HG002, 10 of 10 appeared to be true *de novo*. After manual inspection of 10 random *de novo* indels, 10 of 10 appeared to be true *de novo* indels in homopolymers or tandem repeats. The violations that were not heterozygous in the son and homozygous reference in both parents fell in a few categories: (1) clusters of variants in segmental duplications where a variant was missed or incorrectly genotyped in one individual, (2) complex variants in homopolymers and tandem repeats that were incorrectly called or genotyped in one individual, and (3) some overlapping

complex variants in the MHC that were correctly called in the trio but the different representations were not reconciled by our method (even though we used a method that is robust to most differences in representation).[4,25] We exclude all Mendelian inconsistencies that are not heterozygous in the son and homozygous reference in both parents from the v.4.2.1 benchmark regions of each member of the trio because most are unlikely to have a biological origin. Conservative paternal|maternal phasing for HG002 on GRCh38 was performed for the MHC using local diploid assembly and outside the MHC using phasing that was consistent between trio analysis and integrated Strand-seq and PacBio HiFi phasing (1,812,845 of 2,449,937 heterozygous benchmark variants).

## Regions excluded from the benchmark

A critical part of forming a reliable v.4.2.1 benchmark was to identify regions that should be excluded from the benchmark. In Table 2 and Figure S3, we detail each region type that is excluded, the size of the regions, and reasons for exclusion. We describe how each region is defined in STAR Methods. These excluded regions fall into several categories: (1) the modeled centromere and heterochromatin in GRCh38 because these are highly repetitive regions and generally differ in structure and copy number between any individual and the reference; (2) the VDJ, which encodes immune system components and undergoes somatic recombination in B cells; (3) in GRCh37, regions that are expanded or collapsed relative to GRCh38; (4) segmental duplications with more than 5 copies longer than 10 kb and identity greater than 99%, where errors are likely in mapping and variant calling (e.g., because of structural or copy number variation resulting in calling paralogous sequence variants);[26,27] (5) potential large duplications that are in HG002 relative to GRCh37 or GRCh38; (6) putative insertions, deletions, and inversions greater than 49 bp in size and flanking sequence; and (7) tandem repeats larger than 10,000 bp where variants can be difficult to detect accurately given the length of PacBio HiFi reads. As an example of the importance of carefully

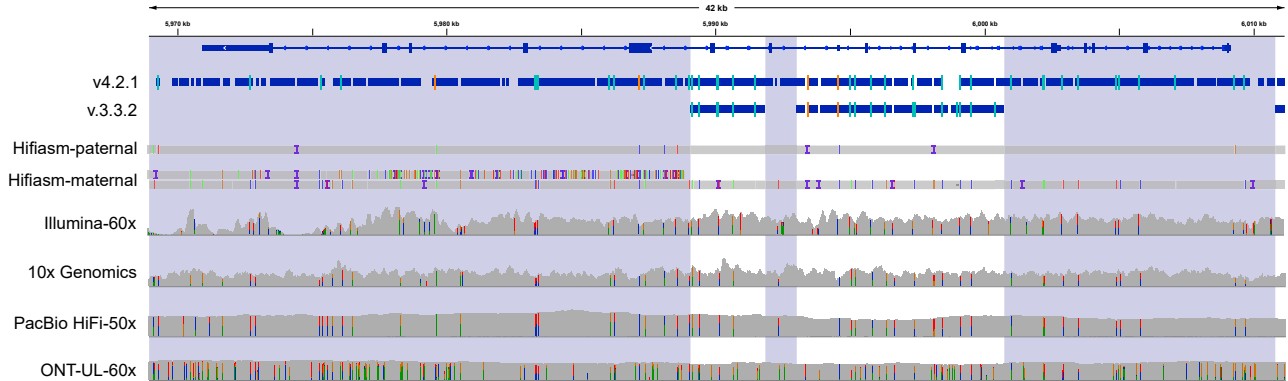

**Figure 4. The difficult-to-map, medically relevant gene *PMS2* is better included in v.4.2.1**
The medically relevant gene *PMS2* is 85.6% included in the v.4.2.1 benchmark regions, whereas it is 25.9% included in v.3.3.2 because segmental duplications (shaded regions) were largely excluded in previous benchmark versions. Thick blue bars indicate regions included by each benchmark, and orange and light blue lines indicate positions of homozygous and heterozygous benchmark variants, respectively. This region is challenging for assembly-based approaches, and an extra contig from the maternal trio-based HG002 Hifiasm assembly (Hifiasm-maternal) aligned to the left half of the gene because of misalignment or misassembly.

excluding questionable regions, when comparing variants from ultralong reads with v.3.3.2, 74% of the putative FPs in HG002 on GRCh38 fell outside the v.4.2.1 benchmark regions (Tables S8 and S9). Many of these were in centromere regions that have very few benchmark variants but were erroneously included in the v.3.3.2 short-read-based benchmark; e.g., in chromosome 20 (chr20). Our new benchmark correctly excludes these regions from the benchmark because they cannot be confidently mapped with short, linked, or long reads used to form the benchmark. Table S10 describes refinements to these excluded regions between the initial draft release and the v.4.2.1 benchmark.

**Evaluation and manual curation demonstrates reliability of the benchmark**
GIAB has established a community evaluation process for draft benchmarks before the official release, following the reliable identification of errors (RIDE) principle for benchmarks.[3] The RIDE principle is designed to ensure that, when comparing state-of-the-art query call sets with the benchmark, at least 50% of the putative false positives and false negatives are errors in the query call set and not errors in the benchmark. GIAB recruited volunteer experts in particular variant calling methods to follow the GA4GH Benchmarking Team's best practices[5] to compare a variety of query variant call sets with the draft benchmarks. We performed the community evaluation on v.4.1 for HG002. Based on this evaluation, we made small improvements to generate v.4.2.1 for HG002 as well as for the other 6 samples (Table S10). v.4.2.1 is the version described in the rest of this manuscript for all samples.

Query call sets for the final evaluation performed on v.4.1 represented a broad range of sequencing technologies and bioinformatics methods (Table S11; STAR Methods). Each call set developer curated a random selection of FPs and FNs to ensure that the benchmark reliably identifies errors in the query call set. Overall, we found that the benchmark was correct and the query call set was not correct in the majority of FP and FN SNVs and

indels (Figure 5, with all curations in Table S12). Overall, 433 of 452 (96%) curated FP and FN SNVs and indels inside v.3.3.2 benchmark regions and 422 of 463 (91%) outside v.3.3.2 benchmark regions were determined to be correct in the v.4.1 benchmark. Some technologies/variant callers, particularly deep-learning-based variant callers using HiFi data, had more sites where it was unclear whether the benchmark was correct or the query call set was correct. These sites tended to be near regions with complex structural variation or places that appeared to be inside potential large duplications in HG002 but were not identified in our CNV approaches. In general, most sites that were not clearly correct in the benchmark and wrong in the query were in regions where the answer was unclear with current technologies (Figure 5B). For example, the v.4.1 benchmark correctly excludes much of the questionable region in Figure S4 but still includes some small regions where there may be a duplication and the variant calls in the benchmark and the query are questionable. Future work will be aimed at developing a new benchmark in the small fraction of questionable regions, but these evaluations demonstrate that the new benchmark reliably identifies FPs and FNs across a large variety of variant call sets, including those based on short, linked, and long reads, as well as mapping-based, graph-based, and assembly-based variant callers.

**New benchmark regions are enriched for false negatives**
We demonstrate the benchmarking utility of v.4.2.1 by comparing an example query call set to the new and old benchmark sets for HG002. For a standard short-read-based call set (Ill GATK-BWA in Figure 5), the number of SNVs missed (even when including filtered variants) was 8.5 times higher when benchmarking against v.4.2.1 than against v.3.3.2 (16,615 versus 1,960). The difference is largely due to false negative SNVs in regions of low mappability and segmental duplications with 15,220 in v.4.2.1 versus 1,381 in v.3.3.2. When counting conservatively filtered SNVs as false negatives, v.4.2.1 detected 71,165 more

**Table 2. Base pairs overlapping different types of difficult regions that are excluded from all input call sets for HG002**

| Difficult region description | Bases excluded in GRCh37 | Bases excluded in GRCh38 | Explanation of exclusion |
| --- | --- | --- | --- |
| Modeled centromere and heterochromatin | N/A | 58,270,517 | highly repetitive regions with modeled reference sequences that are difficult to characterize and structurally variable |
| VDJ | 3,482,644 | 3,348,717 | a region that undergoes somatic recombination |
| Regions that are collapsed and expanded from GRCh37/38 primary assembly alignments | 17,702,248 | N/A | regions of GRCh37 with identified issues, so benchmark small variant calls are generally not as reliable |
| Segmental duplications with >5 copies, >99% identity, and longer than 10 kb | 1,026,737 | 2,094,143 | highly similar duplications with many copies in the reference make it difficult to identify which segmental duplication is the correct location for small variants, and variants could be from structural variants or additional copies of the sequence in HG002 not in the reference |
| Potential increased copy number in HG002 | 21,595,779 | 28,679,205 | difficult to identify in which copy of region the small variants are, could be at a location in GRCh37/38 or at the extra copy of the region in HG002; no standards for representation or benchmarking in these regions |
| Inversions | 843,244 | 893,369 | would need to have a joint small and structural variant benchmark for reliable benchmarking |
| v.0.6 GIAB tier 1 plus tier 2 SV benchmark expanded by 150% | 39,371,460 | 39,560,707 | would need to have a joint small and structural variant benchmark for reliable benchmarking |
| Tandem repeats >10 kb | 1,736,692 | 4,486,559 | these repeats are similar to or longer than the read lengths for all input datasets, making variant calls less reliable |

The table shows progressive subtraction of other difficult regions, so each row has all rows above it subtracted before calculating overlapping base pairs. In non-gap regions on chromosomes 1–22, there are 158,845,257 bp in GRCh37 and 202,943,679 bp in GRCh38 that are excluded by v.4.2.1 (i.e., outside the v.4.2.1 benchmark regions).

errors (183,568 in v.4.2.1 versus 112,403 in v.3.3.2), similar to the increases seen with the noisy long-read-based Syndip benchmark relative to v.3.3.2.[16] Also similar to Syndip, the number of false positive SNVs was higher for v.4.2.1 (25,328) than v.3.3.2 (13,788) before conservative filtering. However, the number of false positive SNVs was actually lower for v.4.2.1 (1,539) than v.3.3.2 (2,370) after conservative filtering, likely because of removal of potential structural and copy number variants in v.4.2.1. Relative to Syndip, v.4.2.1 for HG002 covers about 1% fewer autosomal bases in GRCh38 but 16% more bases in regions of low mappability and segmental duplications. Comparison of the results from the first and second precisionFDA challenges (based on v.3.2 and v.4.2, respectively) demonstrated similar changes in performance when expanding the benchmark; the combined false positive and false negative rates for SNVs increased by 2- to 10-fold when the five top performers of the first challenge were benchmarked against v.4.2.[22] The more challenging variants and regions included in v.4.2.1 enable further optimization and development of variant callers in segmental duplications and low-mappability regions.

## DISCUSSION

We present the first diploid small variant benchmark that uses short, linked, and long reads to confidently characterize a broad spectrum of genomic contexts, including non-repetitive regions as well as repetitive regions, such as many segmental duplications, difficult-to-map regions, homopolymers, and tandem repeats. We demonstrated that the benchmark reliably identifies false positives and false negatives in more challenging regions across many short-, linked-, and long-read technologies and variant callers based on traditional methods, deep learning,[8,9] graph-based references,[10] and diploid assembly.[12] The benchmark was used in the precisionFDA Truth Challenge V2 held in 2020. This challenge focused on difficult regions not covered well by the v.3.2 benchmark used in the first Truth Challenge in 2016, and SNV error rates of the winners of the first Truth Challenge increase by as much as 10-fold when evaluated against the v.4.2 benchmark compared with the v.3.2 benchmark.[22]

We designed this benchmark to cover as much of the human genome as possible with current technologies as long as the benchmark genome sequence is structurally similar to the

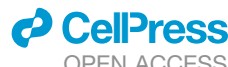

**Figure 5. Summary of manual curations from the evaluation of the v.4.1 benchmark, demonstrating that it reliably identifies FPs and FNs in 13 call sets from different technologies and variant callers**

(A) For each call set, we curated 20 FPs and 20 FNs, and this shows the proportion of curated FP and FN variants where the benchmark set was correct, and the query call set was incorrect. The dashed black line indicates the desired majority threshold, 50%. Half of the curated variants were from GRCh37, and half were from GRCh38.

(B) Breakdown of the total number of variants by category determined during manual curation, where the benchmark curation bar indicates whether the benchmark variant and genotype were determined to be correct and the query curations color indicates whether the query variant and genotype were determined to be correct.

(A and B) Excluded in (B) are variants from (A), where the benchmark was deemed correct and the query incorrect and shows that most of these sites were difficult to curate.

(C) Benchmark-unsure variants by call set. ONT, Oxford Nanopore; PB, PacBio HiFi; Ill, Illumina PCR-free; 10X, 10X Genomics.

GRCh37 or GRCh38 reference. As a linear reference-based benchmark, it has advantages over global *de novo* assembly-based approaches by using reference information to resolve highly homozygous regions and some of the segmental duplications and other repeats where our samples are similar to the reference assembly. This reference-based approach enables users to take advantage of the suite of benchmarking tools developed by the GA4GH Benchmarking Team, including

sophisticated comparison of complex variants, standardized performance metrics, and stratification by variant type and genome context.[5] However, our approach necessitates carefully excluding regions where our reference samples differ structurally from GRCh37 or GRCh38 because of errors in the reference, gaps in the reference, CNVs, or SVs. Developing benchmarks in these regions will require development of methods to characterize these regions with confidence (e.g., using diploid

assembly), standards for representing variants in these regions, and benchmarking methodology and tools. For example, for variants inside segmental duplications for which the individual has more copies than the reference, methods are actively being developed to assemble these regions,[26] but no standards exist for representing on which copy the variants fall or how to compare with a benchmark.

We expect that future benchmarks will increasingly use highly contiguous diploid assembly to access the full range of genomic variation. Our current benchmark is helping to enable this transition by identifying opportunities to improve assemblies in the genome regions that are structurally similar to GRCh37 and GRCh38.

## Limitations of the study

It is critical to understand the limitations of any benchmark. Because our current benchmark excludes regions that structurally differ from GRCh37 or GRCh38, it will not identify deficiencies in mapping-based approaches because of their reliance on these references or highlight advantages of assembly-based approaches that do not rely on these references. Although we have tried to exclude all regions where our samples differ structurally from the reference, some regions with gains in copy number and other large structural variants remain, particularly in segmental duplications where these are more challenging to identify. Similarly, we may not exclude all inversions, particularly those mediated by segmental duplications. In addition, the benchmark still excludes many indels between 15 and 50 bp in size. We also do not characterize mosaic variants, so methods detecting somatic or mosaic variants may identify true variants missing from the benchmark. Although we have significantly increased our inclusion of difficult-to-map, medically relevant genes, more work remains. Many of these genes are excluded because of putative SVs or copy number gains, and future work will be needed to understand whether these are true SVs or copy number gains, and if so, how to fully characterize these regions. The genomes characterized in this work come from individuals of European, Ashkenazi Jewish, and Han Chinese ancestry, and future benchmarks are needed to understand any potential differences in variant benchmarking for other ancestries.

## STAR★METHODS

Detailed methods are provided in the online version of this paper and include the following:

- KEY RESOURCES TABLE
- RESOURCE AVAILABILITY
  ○ Lead contact
  ○ Materials availability
  ○ Data and code availability
- EXPERIMENTAL MODEL AND SUBJECT DETAILS
- METHOD DETAILS
  ○ Incorporating 10x genomics and PacBio HiFi reads into small variant integration pipeline
  ○ Generating callable files with haplotype-separated BAMs
  ○ Python integration

- ○ Regions excluded from the benchmark
- ○ Regions excluded for specific technologies
- ○ Comparing v3.3.2 to v4.2.1
- ○ Calculating difficult-to-map, medically-relevant genes coverage
- ○ Evaluation of the benchmark
- ○ Variant callsets used in evaluation
- ○ Illumina TruSeq DNA PCR-Free reads with VG alignment and Illumina Dragen Bio-IT platform
- ○ Long range PCR confirmation
- ○ Phasing variant calls
- QUANTIFICATION AND STATISTICAL ANALYSIS

## SUPPLEMENTAL INFORMATION

## ACKNOWLEDGMENTS

We thank the Genome in a Bottle Consortium for ongoing feedback and discussions about the benchmark. We thank participants in the precisionFDA Truth Challenge V2 for helpful feedback about the v.4.2 benchmarks for the trio. We thank Valerie Schneider for advice regarding alignments of GRCh38 to GRCh37. Chunlin Xiao was supported by the Intramural Research Program of the National Library of Medicine, National Institutes of Health. P.E. and T.M. acknowledge computational infrastructure provided by the Center for Information and Media Technology at the University of Düsseldorf and funding from the German Research Foundation (grants 391137747 and 395192176) as well as support by the BMBF-funded de.NBI Cloud within the German Network for Bioinformatics Infrastructure (de.NBI) (031A537B, 031A533A, 031A538A, 031A533B, 031A535A, 031A537C, 031A534A, and 031A532B). Certain commercial equipment, instruments, or materials are identified to specify adequately experimental conditions or reported results. Such identification does not imply recommendation or endorsement by the National Institute of Standards and Technology, nor does it imply that the equipment, instruments, or materials identified are necessarily the best available for the purpose.

## AUTHOR CONTRIBUTIONS

Conceptualization, J.W., N.D.O., M.S., and J.M.Z.; data curation, J.W. and N.D.O.; formal analysis – benchmark, J.W., N.D.O., and J.M.Z.; formal analysis – phasing, J.W., P.E., T.M., P.L., V.H., C.-A.M., and J.M.Z.; methodology, J.W., A.C., and J.M.Z.; project administration, J.W. and J.M.Z.; resources, C.X.; software, J.W.; supervision, J.M.Z.; validation, J.W., L.H., Z.K., J.F., M.M., A.S., V.K., A.M.W., W.J.R., C.X., B.Y., N.M., B.N., S.Z., M.K., S.A., M.C.S., G.N., M.B.-B., W.C., U.S.E., C.M., K.S., X.Z., A.S., V.B., A.M.B., I.T.F., A.F., C.-S.C., F.J.S., A.C., and J.M.Z.; visualization, J.W. and N.D.O.; writing – original draft, J.W. and J.M.Z.; writing – review & editing, N.D.O., J.A.R., and F.J.S.

## DECLARATION OF INTERESTS

A.M.W. and W.J.R. are employees and shareholders of Pacific Biosciences. A.M.B. and I.T.F. were employees and shareholders of 10X Genomics. F.J.S. has received sponsored travel from Oxford Nanopore and Pacific Biosciences and a 2018 sequencing grant from Pacific Biosciences. A.S. and V.K. are employees of Seven Bridges. A.C. is an employee of Google Inc. and a former employee of DNAnexus. A.F. and C.-S.C. are employees of DNAnexus.

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

## STAR★METHODS

### KEY RESOURCES TABLE

| REAGENT or RESOURCE | SOURCE | IDENTIFIER |
|---|---|---|
| **Deposited data** | | |
| Extensive data for GIAB samples are available at the NCBI SRA in the GIAB BioProject | GIAB | NCBI BioProject: PRJNA200694 |
| Complete Genomics paired end 26 × 26 bp, ∼100× coverage, processed by Complete Genomics pipeline | GIAB | NCBI SRA: SRX852932 to SRX852936 |
| Illumina PCR-free paired end 150 × 150 bp, ∼300× coverage for HG001-HG004 and ∼100× coverage for HG006 and HG007 | GIAB | NCBI SRA: SRX1049768 to SRX1049855, SRX847862 to SRX848317 |
| Illumina PCR-free paired end 250 × 250 bp, ∼300× coverage for HG005 and ∼40-50× coverage for HG001-HG004 | GIAB | NCBI SRA: SRX1388368 to SRX1388459, SRX1726841 to SRX1726859, SRX1726861 to SRX1726869 |
| Illumina 6 kb mate-pair, ∼13× coverage for HG002-HG007 | GIAB | NCBI SRA: SRX1388732 to SRX1388743 |
| PacBio HiFi/CCS Sequel II | Wenger et al 2020 | NCBI SRA: SRX7083054 to SRX7083059 |
| **Experimental models: Cell lines** | | |
| Mother from CEPH Utah Pedigree (HG001) | NIST Office of Reference Materials; Coriell/NIGMS | NIST RM8398; GM12878; RRID: CVCL_7526 |
| Son of Ashkenazi Jewish ancestry (HG002) | NIST Office of Reference Materials; Coriell/NIGMS; PGP | NIST RM8391/RM8392; GM24385; RRID: CVCL_1C78 |
| Father of Ashkenazi Jewish ancestry (HG003) | NIST Office of Reference Materials; Coriell/NIGMS; PGP | NIST RM8392; GM24149; RRID: CVCL_1C54 |
| Mother of Ashkenazi Jewish ancestry (HG004) | NIST Office of Reference Materials; Coriell/NIGMS; PGP | NIST RM8392; GM24143; RRID: CVCL_1C48 |
| Son of Han Chinese ancestry (HG005) | NIST Office of Reference Materials; Coriell/NIGMS; PGP | NIST RM8393; GM24631; RRID: CVCL_1C97 |
| Father of Han Chinese ancestry (HG006) | Coriell/NIGMS; PGP | GM24694; RRID: CVCL_1C98 |
| Mother of Han Chinese ancestry (HG007) | Coriell/NIGMS; PGP | GM24695; RRID: CVCL_1C99 |
| **Software and algorithms** | | |
| Code used to integrate data and form benchmark | This manuscript | Zenodo: https://doi.org/10.5281/zenodo.5907973 |
| BWA MEM | | https://github.com/lh3/bwa |
| GATK | | https://gatk.broadinstitute.org/hc/en-us |
| freebayes | | https://github.com/freebayes/freebayes |
| LongRanger | | https://github.com/10XGenomics/longranger |
| DeepVariant | | https://github.com/google/deepvariant |
| whatshap | | https://github.com/whatshap/whatshap/tree/bb7ccfffc655072451d642b4eea9661f96b345af |
| mosdepth | | https://github.com/brentp/mosdepth |
| mrCaNaVar | | https://github.com/BilkentCompGen/mrcanavar |
| SVRefine | | https://github.com/nhansen/SVanalyzer |
| TrioCanu | | https://github.com/marbl/canu |
| Flye | | https://github.com/fenderglass/Flye |
| bedtools | | https://github.com/arq5x/bedtools2 |
| pbsv | | https://github.com/PacificBiosciences/pbsv |
| hap.py | | https://github.com/Illumina/hap.py |

| *Continued* | | |
|---|---|---|
| REAGENT or RESOURCE | SOURCE | IDENTIFIER |
| Other | | |
| Sequence data, analyses, and resources related to the NIST Genome in a Bottle Consortium samples in this manuscript | This paper | https://www.nist.gov/programs-projects/genome-bottle |
| GIAB stratifications used for benchmarking | | NIST Public Data Repository: https://doi.org/10.18434/mds2-2499 |
| v4.2.1 Benchmark VCF and BED files | This paper | https://ftp-trace.ncbi.nlm.nih.gov/ReferenceSamples/giab/release/ |

## RESOURCE AVAILABILITY

### Lead contact
Further information and requests for resources and reagents should be directed to and will be fulfilled by the lead contact, Justin Zook (justin.zook@nist.gov).

### Materials availability
DNA extracted from a single large batch of cells for 5 of the 7 genomes (HG001 to HG005) is publicly available in National Institute of Standards and Technology Reference Materials 8391 (HG002), 8392 (HG002-HG004), 8393 (HG005), and 8398 (HG001), available at https://www.nist.gov/srm. DNA for HG001 to HG005, as well as HG006 and HG007, are extracted from publicly available cell lines GM12878 (HG001, RRID:CVCL_7526), GM24385 (HG002, RRID:CVCL_1C78), GM24149 (HG003, RRID:CVCL_1C54), GM24143 (HG004, RRID:CVCL_1C48), GM24631 (HG005, RRID:CVCL_1C97), GM24694 (HG006, RRID:CVCL_1C98), and GM24695 (HG007, RRID:CVCL_1C99) at the Coriell Institute for Medical Research National Institute for General Medical Sciences cell line repository.

### Data and code availability
- DNA sequencing data were previously deposited in the NCBI SRA under BioProject PRJNA200694 and are publicly available as of the date of publication. Accession numbers are listed in the key resources table.
- All original code has been deposited at Zenodo and is publicly available as of the date of publication. DOIs are listed in the key resources table.
- Any additional information required to reanalyze the data reported in this paper is available from the lead contact upon request.

## EXPERIMENTAL MODEL AND SUBJECT DETAILS

The Genome in a Bottle Consortium selected the seven human lymphoblastoid cell lines described in Materials availability for characterization because the pilot female HG001 had extensive pre-existing public data, and HG002 to HG007 are two son-father-mother trios from the Personal Genome Project that have a broader consent that permits commercial redistribution and recontacting participants for further sample collection.

## METHOD DETAILS

### Incorporating 10x genomics and PacBio HiFi reads into small variant integration pipeline
v4.2.1 uses the same variant call sets as v3.3.2 from Complete Genomics,[28] Illumina PCR-free (novoalign, GATK, and freebayes), and Illumina mate-pair (bwa mem, GATK, and freebayes).[29–31] v4.2.1 uses 10x Genomics linked-read data and the variant calls from the LongRanger pipeline,[6] which makes calls both with and without using information from partitioning reads into haplotypes. In v3.3.2, we used conservative, haplotype-separated GATK calls from 10x Genomics, where calls were only made separately on each haplotype and coverage from both haplotypes was required. Also, v4.2.1 uses PacBio HiFi data using Sequel II with read lengths of 15 kb and 20 kb merged into a dataset that has approximately 47x to 68x coverage (Table S1), with variants subsequently called by GATK4 and DeepVariant.[7,8] The 10x and PacBio HiFi data provide access to genomic regions that were previously inaccessible to short reads including segmental duplications. As shown in Table 1 the number of base pairs in the benchmark that covers segmental duplications has increased with the incorporation of long- and linked-read data. Key Resources Table lists the input datasets for the small variant integration pipeline to produce v4.2.1 for HG002 from 10x Genomics, Complete Genomics, Illumina 150 bp, Illumina 250 bp, Illumina 6 kbp mate-pair, and PacBio Sequel II HiFi/CCS.

### Generating callable files with haplotype-separated BAMs

We use the CallableLoci utility in GATK3 to find regions with good coverage of high mapping quality reads. For PacBio HiFi and 10x Genomics read data, we use WhatsHap[32] haplotag to partition reads by haplotype then use the bamtools filter function to generate separate BAM files for the two haplotypes. To partition reads by haplotype, we used a vcf that combined 10x linked read phasing with trio information described in the v0.6 structural variant benchmark paper.[3] For CallableLoci with the unseparated BAM, we set the callable maxDepth threshold to 2 times the median coverage for VCF entries, then the minDepth threshold to 20. For the haplotype separated BAMs, we use median coverage for VCF as the maxDepth and 5 as the minDepth.

For PacBio HiFi, we first remove multi-allelic entries from the VCF and 50 bp on each side of the variant then remove RefCall entries with QUAL value below 40 along with 50 bp on each side of those variants. We then find callable regions for each haplotype BAM and the unseparated BAM then use bedtools multiIntersectBed to find the union of those regions.

For 10x Genomics, we first remove filtered indels along with 50 bp on each side from its callable regions. Then we find callable regions on each haplotype and the unseparated BAM. After using multiIntersectBed to find the union of those callable regions we subtract regions that were callable on one haplotype but not callable on the other haplotype.

### Python integration

We implemented the integration pipeline using Python as opposed to the Bash and Perl implementation for v3.3.2. The integration maintains a similar structure and we generated a DNAnexus applet to run on the same platform as v3.3.2. We updated the v4.2.1 pipeline to exclude all of a tandem repeat that is only partially covered by the benchmark regions. We also provide an option to not provide a callable file for given callsets, which for v4.2.1 we do not use callable regions for Ion Torrent or SOLiD. This results in a benchmark VCF that includes annotations for those technologies but variants are not excluded based on disagreement with their calls.

### Regions excluded from the benchmark

We determined regions to exclude from the benchmark where it was not currently possible to determine which technologies were correct due to the difficulty of resolving variation in that region. The difficult regions included those that had a structural variant identified in the GIAB SV v0.6 Benchmark, regions in which the HG002 sample had a copy variation compared to the reference, high depth and highly similar segmental duplications, tandem repeats >10 kb, regions that are collapsed and expanded from GRCh37/38 Primary Assembly Alignments, modeled centromere and heterochromatin, VDJ, and inversions. The bed files excluded from the benchmark are being made available in the v3.00 stratifications from GIAB under https://ftp-trace.ncbi.nlm.nih.gov/ReferenceSamples/giab/release/genome-stratifications/. We refined these regions with several rounds of internal and external evaluation on intermediate versions of the benchmark. We describe intermediate versions of the benchmark in Table S10.

#### Modeled centromere and heterochromatin

We use the modeled centromere for GRCh38 from ftp://ftp-trace.ncbi.nlm.nih.gov/ReferenceSamples/giab/release/NA12878_HG001/NISTv3.3.2/GRCh38/supplementaryFiles/genomic_regions_definitions_modeledcentromere.bed and the heterochromatin region ftp://ftp-trace.ncbi.nlm.nih.gov/ReferenceSamples/giab/release/NA12878_HG001/NISTv3.3.2/GRCh38/supplementaryFiles/genomic_regions_definitions_heterochrom.bed.[33]

#### VDJ region subject to somatic recombination

We downloaded the UCSC genes tracks[34] from http://hgdownload.cse.ucsc.edu/goldenPath/hg19/database/kgXref.txt.gz and selected entries with "abParts". We then subset to chromosomes 2, 14, and 22 which contain the IGK, IGH, and IGL components that make up the VDJ recombination regions.

#### KIR region

v4.2.1 excludes the KIR region (chr19:54716689-54871732 in GRCh38 and 19:55228188-55383188 in GRCh37) because it is highly variable in copy number in the population, variant representation is challenging, and our current mapping-based methods have more errors in this region.

#### Regions that are collapsed and expanded from GRCh37/38 primary assembly alignments

The GRC aligned GRCh37 to GRCh38 (excluding alts) with results available at: ftp://ftp.ncbi.nlm.nih.gov/pub/remap/Homo_sapiens/2.1/GCF_000001405.13_GRCh37/GCF_000001405.26_GRCh38/. We parsed the file GCF_000001405.13.xlxs for two Discrepancy values: (1) SP that denotes collapsed regions and (2) SP-only that denotes a region that was expanded between the reference versions.

#### Highly similar and high depth segmental duplications longer than 10kb

We begin with the segmental duplications track from UCSC:[34] http://hgdownload.cse.ucsc.edu/goldenPath/hg19/database/genomicSuperDups.txt.gz. We filter for entries larger than 10 kb and with identity >99%. We then use bedtools genomecov to calculate segmental depth and subset to those greater than 5.

#### Potential copy number variation

We employed several approaches to find potential regions of large duplications in HG002 that are not in GRCh37 and GRCh38:

1. Short read and Long Read Intersection: We used mosdepth[35] to find 1,000 bp windows that have higher than average coverage/2*2.5 in ONT and PacBio HiFi data. We intersected these regions with results from the CNV analysis tool, mrCaNaVar,[36] on Illumina

HiSeq 300x data (ftp://ftp-trace.ncbi.nlm.nih.gov/ReferenceSamples/giab/data/AshkenazimTrio/analysis/BilkentUni_IlluminaHiSeq_TARDIS_mrCaNaVar_05212019/AJtrio-HG002.hs37d5.300x.bam.bilkentuniv.052119.dups.bed.gz and ftp://ftp-trace.ncbi.nlm.nih.gov/ReferenceSamples/giab/data/AshkenazimTrio/analysis/BilkentUni_mrCaNaVaR_GRCh38_07242019/AJtrio-HG002.hg38.300x.bam.bilkentuniv.072319.dups.bed.gz).

2  Diploid Assemblies of HG002: We used SVRefine (https://github.com/nhansen/SVanalyzer) to align diploid assemblies to GRCh37/GRCh38 with bedgraph files that denote coverage of the reference by the number of contigs for the maternal and paternal haplotypes. We used bedtools unionBedGraphs and then found reference regions that are covered by 2 or more contigs in the union of haplotypes. We did this separately on a TrioCanu assembly using ONT,[37] a Flye assembly using ONT,[38] and a TrioCanu assembly of PacBio HiFi 15 kb reads.[7] We found an intersection across the three assemblies and subset to regions greater than 10 kb.

3  Elliptical Outlier Boundary with PacBio HiFi and ONT sequencing data: We used mosdepth to calculate coverage in 1,000 bp windows of the PacBio HiFi data and the ONT ultralong dataset (ftp://ftp-trace.ncbi.nlm.nih.gov/ReferenceSamples/giab/data/AshkenazimTrio/HG002_NA24385_son/Ultralong_OxfordNanopore/guppy-V2.3.4_2019-06-26/ultra-long-ont_hs37d5_phased.bam and ftp://ftp-trace.ncbi.nlm.nih.gov/ReferenceSamples/giab/data/AshkenazimTrio/HG002_NA24385_son/Ultralong_OxfordNanopore/guppy-V2.3.4_2019-06-26/ultra-long-ont_GRCh38_reheader.bam). We then found regions that had outlier coverage in PacBio HiFi and/or ONT. To do so, as described in the equations below, we (1) divided the PacBio HiFi coverage of each window by the median depth HiFi depth and squared it; (2) divided the ONT coverage of each window by the median depth ONT depth and squared it; (3) summed those values; and (4) took the square root of the sum. We found the third quartile and interquartile range for those transformed window coverage values. Finally, we found windows with coverage greater than the third quartile plus 1.5 times the IQR. In the equations below, WindowHiFiDepth, WindowONTDepth, and EllipticalValues are vectors, while MedianHiFiDepth, MedianONTDepth, and ThresholdEllipticalOutlier are scalars.

$$windowHifiDepth = \text{mean HiFi read depth in 1000bp window from mosdepth}$$
$$windowONTDepth = \text{mean ONT read depth in 1000bp window from mosdepth}$$
$$MedianHifiDepth = median(WindowHifiDepth)$$
$$MedianONTDepth = median(WindowONTDepth)$$
$$EllipticalValues = \sqrt{\left(\left((WindowHiFiDepth/medianHiFiDepth)^2 + (WindowONTDepth/medianONTDepth)^2\right)\Big/2\right)}$$
$$ThresholdEllipcticalOutlier = percentile(EllipticalValues, 75) + 1.5 * IQR(EllipticalValues)$$

### Inversions

We used SVrefine (github commit f0fb99969b6e239d1f49bc64a8f6cf.5d52a2b88b) to call structural variants with, –maxsize 1000000 option. We then extracted inversions from the call set. Variants were merged with SVmerge (github commit aa8beb6f1cb5c539eea9f980ff30d2648caeee21), default maximum "distances", which were 0.2 for all. SVrefine and SVmerge were from SVanalyzer (https://github.com/nhansen/SVanalyzer).

### HG002 v0.6 GIAB Tier1 plus Tier 2 SV Benchmark expanded by 150%

We used bedtools[39] slop with parameters -b -pct 0.25 to expand the GIAB v0.6 Structural Variant benchmark file: ftp://ftp-trace.ncbi.nlm.nih.gov/ReferenceSamples/giab/data/AshkenazimTrio/analysis/NIST_SVs_Integration_v0.6/HG002_SVs_Tier1plusTier2_v0.6.1.bed. This file defines regions in which calls with PASS in the FILTER field as well as regions should contain close to 100% of true insertions and deletions ≥ 50 bp, with variants merged into a single region if they were within 1 kb.

### SVs excluded from HG001 and HG003-HG007

Because we don't have SV benchmarks for HG001 and HG003-HG007, we used pbsv (https://github.com/PacificBiosciences/pbsv) SVs >49 bp from PacBio HiFi data for HG001 and HG003-HG007, and well as svanalyzer and dipcall SVs >49 bp from trio-hifiasm assemblies of HG001 and HG005. We expanded these SVs to include overlapping tandem repeats and homopolymers and expanded the resulting regions by 25% of the region size on each side with bedtools[39] slop with parameters -b -pct 0.25.

### Tandem repeats greater than 10,000 bp

We took the union of SimpleRepeat dinucleotide, trinucleotide, and tetranucleotide STRs as well as RepeatMasker_LowComplexity, RepeatMasker_SimpleRepeats, and TRF_SimpleRepeats downloaded from UCSC Genome Browser. We then subset to tandem repeats longer than 10,000 bp.

### Reference assembly contigs shorter than 500,000 bp

We downloaded the gap track from UCSC:[34] ftp://hgdownload.cse.ucsc.edu/goldenPath/hg19/database/gap.txt.gz. Then subset to regions that are gap. We used bedtools complemented with GRCh37/GRCh38 to find contigs then subset to those less than 500 kb.

### Regions excluded for specific technologies

We exclude tandem repeats approximately larger than the read length from each method. Tandem repeats shorter than 51 bp were excluded from all technologies except Illumina PCR-free GATK, Complete Genomics, and PacBio HiFi DeepVariant. We excluded tandem repeats between 51 bp and 200 bp except for Illumina PCR-Free GATK and PacBio HiFi DeepVariant. Tandem

repeats between 200 bp and 10,000 bp are excluded from all technologies except PacBio HiFi DeepVariant. Homopolymers greater than 6 bp were excluded from all technologies except Illumina PCR-free GATK, Complete Genomics, Ion Exome, PacBio HiFi DeepVariant. Imperfect homopolymers greater than 10 bp were excluded from all technologies except Illumina PCR-Free GATK. Low mappability regions that are difficult to map for short reads were excluded from all except 10x Genomics and PacBio HiFi. LINE:L1Hs greater than 500 bp were excluded except Illumina MatePair, 10x Genomics, and PacBio HiFi. Segmental duplications were excluded from all technologies except 10x Genomics and PacBio HiFi. Homopolymers were excluded from 10x Genomics and PacBio HiFi. Long homopolymers were included only for GATK based calls for PCR-Free data because GATK gVCF has low genotype quality score if reads do not totally encompass the homopolymer. Overall we trust homopolymers most from PCR-Free short reads. We visualize the regions excluded from each sequencing technology in Figure S5.

### Comparing v3.3.2 to v4.2.1

For HG002, we subset v3.3.2 variants to v3.3.2 benchmark bed and v4.2.1 variants to v4.2.1 benchmark bed and compared the benchmarks using hap.py with v2.0 of the GA4GH benchmarking stratifications (https://github.com/ga4gh/benchmarking-tools).[5] To identify the types of genomic regions where v4.2.1 gains and loses benchmark variants relative to v3.3.2, we subset to stratifications with at least 1000 variants in v4.2.1, and sorted by the difference between the Precision and Recall metrics, which are measures of the fraction of extra variants in v3.3.2 and v4.2.1, respectively.

### Calculating difficult-to-map, medically-relevant genes coverage

We used the 193 clinically-relevant gene names that contained exons that are difficult to map with short reads from.[13] We used Ensembl BioMart to retrieve Human Genes Build 99 with Gene Name, Start, End, and Chromosome (http://jan2020.archive. ensembl.org/biomart/martview/2c3a4b803e1a01b3b806829a466b3590).[40] We used those results to find coordinates for the difficult-to-map, medically-relevant gene names, subset to genes on chromosomes 1-22, then used bedtools intersect with the v3.3.2 and v4.2.1 benchmark region files to find overlap.

### Evaluation of the benchmark

We used hap.py (https://github.com/Illumina/hap.py) following GA4GH best practices[5] with HG002 v4.1 benchmark variants as the truth set, v4.1 benchmark bed as confident regions, and each of the 12 call sets as the query. We use the vcfeval engine for comparison.[25]

To evaluate the utility of the v4.1 benchmark, the GIAB community contributed 13 call sets from short-, linked-, and long-read technologies, and from mapping-, graph-, and assembly-based variant callers. We used hap.py to compare each input callset to v4.1 then asked collaborators to manually curate a small subset of the False Positive and False Negative sites with commands detailed in "Supplementary Materials - Benchmark Evaluations". Collaborators evaluated 5 False Positive SNVs, 5 False Positive Indels, 5 False Negative SNVs, 5 False Negative Indels both inside and outside v3.3.2 along with 5 False Positive SNVs, 5 False Positive Indels, 5 False Negative SNVs, 5 False Negative Indels in the MHC for GRCh37. We generated IGV sessions with BAM files for Illumina HiSeq, 10x Genomics, PacBio HiFi 15 kb & 20 kb merged, and ONT Ultralong,[11] then asked that the evaluators identify for each site if both alleles in the benchmark were correct and if both alleles in the query call set were correct.

### Variant callsets used in evaluation
#### PacBio HiFi reads with GATK Haplotype caller

HG002 HiFi reads from three publicly available datasets were aligned to the GRCh37 and GRCh38 references using the pbmm2 v0.10.0 with '–preset CCS' (https://github.com/PacificBiosciences/pbmm2). The two Sequel I datasets with 10 kb and 15 kb insert sizes are available at ftp://ftp.ncbi.nlm.nih.gov/ReferenceSamples/giab/data/AshkenazimTrio/HG002_NA24385_son/PacBio_CCS_10kb and ftp://ftp.ncbi.nlm.nih.gov/ReferenceSamples/giab/data/AshkenazimTrio/HG002_NA24385_son/PacBio_CCS_15kb, respectively (SRA SRX5327410). The Sequel II dataset with 11 kb insert size is available at ftp://ftp.ncbi.nlm.nih.gov/ReferenceSamples/giab/data/AshkenazimTrio/HG002_NA24385_son/PacBio_SequelII_CCS_11kb (SRA SRX5527202).

Small variants were called with GATK v4.0.10.1 HaplotypeCaller with '–pcr-indel-model AGGRESSIVE' and '–minimum-mapping-quality 10' (https://github.com/broadinstitute/gatk/releases/tag/4.0.10.1). Variants were filtered on the QD (Quality by Depth) value with GATK v4.0.10.1 Variant Filtration, such that:

- SNVs with QD < 2 are filtered
- Indels > 1 bp with QD < 2 are filtered
- 1 bp Indels with QD < 5 are filtered

GRCh37 reference used for alignment: ftp://ftp-trace.ncbi.nih.gov/1000genomes/ftp/technical/reference/phase2_reference_assembly_sequence/hs37d5.fa.gz

GRCh38 reference used for alignment: ftp://ftp.ncbi.nlm.nih.gov/genomes/all/GCA/000/001/405/GCA_000001405.15_GRCh38/seqs_for_alignment_pipelines.ucsc_ids/GCA_000001405.15_GRCh38_no_alt_analysis_set.fna.gz

### PacBio Hifi reads using minimap2 with DeepVariant

A set of ∼80x coverage PacBio CCS data was mapped to each reference:

> minimap2 VN:2.15-r905
> minimap2 -ax asm20 -t 32
> (Note that the mapping of these files predates some improved recommendations for mapping to use pbmm2)

DeepVariant v0.8 with the PACBIO model was applied to the mapped files. The commands and workflow used are identical to the DeepVariant case-study:

https://github.com/google/deepvariant/blob/r0.8/docs/deepvariant-pacbio-model-case-study.md

No filtering is applied.

### PacBio HiFi reads realigned using Duplomap

HG002 HiFi reads aligned to the GRCh37 reference using Minimap2 were downloaded from ftp://ftp-trace.ncbi.nlm.nih.gov/ReferenceSamples/giab/data/AshkenazimTrio/HG002_NA24385_son/PacBio_CCS_15kb_20kb_chemistry2/ and reads overlapping segmental duplications were realigned using a tool Duplomap (https://gitlab.com/tprodanov/duplomap) that used paralogous sequence variants to map reads with multiple possible alignment locations. Small variants were called from the realigned bam file using DeepVariant v.0.8 with default parameters.

### 10x Genomics using Aquila local assembly

Aquila uses linked-read data for generating a high quality diploid genome assembly, from which it then comprehensively detects and phases personal genetic variation. Here, Aquila merged two link-reads libraries to generate WGS variant calls for NA24385. Assemblies and VCFs for this merged library L5 + L6 can be found at http://mendel.stanford.edu/supplementarydata/zhou_aquila_2019/ The raw linked-reads fastq files can be downloaded in the Sequence Read Archive and its BioProject accession number is PRJNA527321.

### Illumina TruSeq DNA PCR-Free reads with Illumina Dragen Bio-IT platform

Illumina PCR-Free reads (2 x 250 bp with 350 bp insert size) are downloaded from the public file server.

Dragen 3.3.7 is used to perform alignment, variant calling, and filtering on GRCh37 and GRCh38 reference assemblies. Variant filtering is based on MQ (Mapping Quality), MQRankSum (Z-score From Wilcoxon rank sum test of Alt vs Ref read MQs), and ReadPosRankSum (Z-score from Wilcoxon rank sum test of Alt vs Ref read position bias) values. For SNVs, $MQ < 30.0$, $MQRankSum < -12.5$, or $ReadPosRankSum < -8.0$ are filtered out. For INDEL, $ReadPosRankSum < -20.0$ are filtered.

### Illumina PCR-Free reads are downloaded from

ftp://ftp-trace.ncbi.nlm.nih.gov/ReferenceSamples/giab/data/AshkenazimTrio/HG002_NA24385_son/NIST_Illumina_2 × 250bps/reads/

### Illumina TruSeq DNA PCR-Free reads with VG alignment and Illumina Dragen Bio-IT platform

Illumina PCR-Free read pairs (2 x 250bp with 350 bp insert size) are downloaded from and extracted from novoaligned bams that are hosted on the public file server. The process is based on aligning the HG002 to genome graphs that were constructed from HG003 and HG004 parental variants. All alignments are performed using Variation Graph Toolkit (VG) and variant calling is done using Dragen version 3.2. Default variant calling settings in Dragen 3.2 were used during GVCF and VCF variant calling. The methods used to convert graph alignments to linear alignments and parental graph construction are in the workflow defined on the vg_wdl GitHub repository.

The workflow used to process this data can be found here https://github.com/vgteam/vg_wdl/blob/master/workflows/vg_trio_multi_map_call.wdl

Illumina PCR-Free reads for the trio used in parental graph construction and HG002 alignment are downloaded from

ftp://ftp-trace.ncbi.nlm.nih.gov/ReferenceSamples/giab/data/AshkenazimTrio/HG002_NA24385_son/NIST_Illumina_2 × 250bps/novoalign_bams/

ftp://ftp-trace.ncbi.nlm.nih.gov/ReferenceSamples/giab/data/AshkenazimTrio/HG003_NA24149_father/NIST_Illumina_2 × 250bps/novoalign_bams/

ftp://ftp-trace.ncbi.nlm.nih.gov/ReferenceSamples/giab/data/AshkenazimTrio/HG004_NA24143_mother/NIST_Illumina_2 × 250bps/novoalign_bams/

The population data used for initial graph alignments of the HG002 trio samples are based on the 1000 genomes phase 3 variant dataset and the GRCh37 reference genome. http://ftp.1000genomes.ebi.ac.uk/vol1/ftp/release/20130502/ALL.wgs.phase3_shapeit2_mvncall_integrated_v5b.20130502.sites.vcf.gz

### 10x genomics using LongRanger with GATK Haplotype Caller

These callsets, generated independently for each individual in the Ashkenazi trio, used LongRanger[21] (version 2.2, code at https://github.com/10XGenomics/longranger) and GATK v4.0.0.0 as variant caller with default parameters on 10x Genomics linked-reads data for the family trio (84x, 70x, and 69x coverage for HG002 NA24385 son, HG003 NA24149 father, and HG004 NA24143 mother, respectively) against both GRCh37 and GRCh38. The vcf and bam files for each genome are under:

ftp://ftp-trace.ncbi.nlm.nih.gov/ReferenceSamples/giab/data/AshkenazimTrio/analysis/10XGenomics_ChromiumGenome_LongRanger2.2_Supernova2.0.1_04122018/

The variant curation used the 10x Genomics VCF from LongRanger 2.2 (SRA accession SRX2225480), which is available at: https://ftp-trace.ncbi.nlm.nih.gov/ReferenceSamples/giab/data/AshkenazimTrio/analysis/10XGenomics_ChromiumGenome_LongRanger2.2_Supernova2.0.1_04122018/GRCh37/NA24385_300G/NA24385.GRCh37.phased_variants.vcf.gz

All samples were sequenced on the Illumina Xten at 2 × 150 bp. The Ashkenazim trio was done using the v1 of the 10x library prep protocol.

### HiFi clair

This callset was generated using Sequel II 11 kbp HiFi reads aligned to the hs37d5 reference with pbmm2, publicly available here: https://ftp-trace.ncbi.nlm.nih.gov/ReferenceSamples/giab/data/AshkenazimTrio/HG002_NA24385_son/PacBio_SequelII_CCS_11kb/. The variants were called by using Clair (v1) on these alignments.

### Illumina Novaseq 2 × 250 bp data

The sample HG002 was sequenced on an Illumina Novaseq 6000 instrument with 2 × 250 bp paired end reads at the New York Genome Center. The libraries were prepped using Truseq DNA PCR-free library preparation kit. The raw reads were aligned to both GRCh37 and GRCh38 human reference. Alignment to GRCh38 reference, marking duplicates and base quality recalibration was performed as outlined in the Centers for Common Disease Genomics (CCDG) functional equivalence paper. Alignment to GRCh37 was performed using BWA-Mem (ver. 0.7.8) and marking duplicates using Picard (ver. 1.83) and local Indel realignment and base quality recalibration using GATK (ver. 3.4–0). Variant calling was performed using GATK (ver. 3.5) adhering to the best practices recommendations from the GATK team. Variant calling constituted generating gVCF using HaplotypeCaller, genotyping using the GenotypeGVCFs subcommand and variant filtering performed using VariantRecalibrator and ApplyRecalibration steps. A tranche cutoff of 99.8 was applied to SNP calls and 99.0 to InDels to determine PASS variants. The raw reads are available for download at SRA at https://www.ncbi.nlm.nih.gov/sra/SRX7925517, https://www.ncbi.nlm.nih.gov/sra/SRX7925518, and https://www.ncbi.nlm.nih.gov/sra/SRX7925519.

### Long range PCR confirmation

We performed Long range PCR followed by Sanger sequencing for variants in LINEs and difficult-to-map, medically-relevant genes for all 7 samples. The difficult genes that were chosen for long-range PCR and Sanger sequence confirmation are potentially medically-relevant and have many characteristics that make them difficult to characterize, especially with short reads. We selected genes with previously published long range PCR assays. The first set of genes make up the RCCX complex, a segmental duplication that includes *TNXA*, *TNXB*, *C4A*, *C4B*, and *CYP21A2*.[41,42] The similar sequences of these genes in close proximity makes them prone to rearrange, mutate and change the size of the complex as a whole, and they are linked to rare diseases that are inherited together at a higher rate than would be expected by chance. Mutations in the *CYP2D6* gene can affect metabolism and bioactivation of many clinical drugs and the gene contains a polymorphic region.[43] *DMBT1* has been identified as a candidate tumor suppressor for brain, gastrointestinal and lung cancers and contains highly repetitive sequence.[44] Rare variants in the *HSPG2* gene are linked to cases of idiopathic scoliosis.[45] *STRC* has a pseudogene with high genomic and coding sequence homology making it very difficult to characterize by normal short read sequencing methods.[46] The *PMS2* gene has multiple pseudogenes, making it difficult to reliably detect mutations or characterize by sequencing.[23] We additionally include v4.2.1 variants covered by the long range PCR assays designed for genes as described for the GIAB Challenging Medically Relevant Gene benchmark.[20]

Long range PCR was performed to amplify regions with variants in LINEs and difficult-to-map, medically-relevant genes. Primers for amplification of LINEs were designed with the Primer3Plus software.[47] Other primers were sourced from literature. All long range primer sequences and references can be found in Table S13. Long range PCR were performed with the PrimerSTAR GXL DNA Polymerase (Takara Bio, Mountain View, CA), and assays specific reaction components can be found in Table S14. Long range PCR conditions varied by assay and can be found in Table S15.

Sanger primers were designed using the Primer3Plus software.[47] Primer sequences can be found in Table S13. Long range PCR products were purified with ExoSAP-IT (Applied Biosystems, Foster City, CA). Sanger sequencing was performed with SimpleSeq Premixed Sequencing Kits (Eurofins Genomics, Louisville, KY) using 5 mL of the long range PCR amplicon and 5 mL of 3 mM primer. Sanger sequencing traces were aligned and analyzed with Geneious Prime (Biomatters, Inc., San Diego, CA).

### Phasing variant calls

To provide initial conservative phasing information for regions including the MHC and segmental duplications, the v4.2.1 benchmark vcf for HG002 on GRCh38 was phased in 3 ways. For the MHC, phasing was obtained from the fully phased local diploid assembly, using trio information to ensure it follows the paternal|maternal convention in the GT field. For the rest of the genome, we used phased heterozygous calls that were consistent in a single phase block for each chromosome between trio-based phasing and integrative phasing using Strand-seq and PacBio HiFi reads. The HG002 v4.2.1 benchmark variants were phased independently from the parental variants using integrative phasing.[48] The integrative phasing approach combined local phase information from PacBio HiFi long-read alignments with global phase information obtained from Strand-seq short-read alignments to create whole-chromosome haplotypes for each individual. Method and implementation

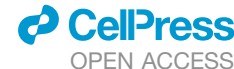

were applied as previously described[49] with minor modifications: the GRCh38 assembly was used as reference for both PacBio HiFi long-read and Strand-seq short-read alignments, and the "–indels" option was added to the "whatshap phase" command line.

Additionally, for the children HG001, HG002, and HG005, we transferred paternal|maternal phasing from a dipcall[16] vcf using a trio-hifiasm v0.11 assembly[50] to v4.2.1 vcf of each individual. These draft phased vcfs, which have not been evaluated for accuracy, are available under the Supplementary Files directory for HG001, HG002, and HG005 at https://ftp-trace.ncbi.nlm.nih.gov/ReferenceSamples/giab/release/.

## QUANTIFICATION AND STATISTICAL ANALYSIS

No statistical analyses were performed in this work.

