## [Document S2. Transparent peer review records for Wagner et al · Cell Genomics]

Benchmarking challenging small variants with linked and long reads

Author list

Justin Wagner,1,* Nathan D Olson,1 Lindsay Harris,1 Ziad Khan,2 Jesse Farek,2 Medhat Mahmoud,2 Ana Stankovic,3 Vladimir Kovacevic,3 Byunggil Yoo,4 Neil Miller,4 Jeffrey A. Rosenfeld,5 Bohan Ni,6 Samantha Zarate,6 Melanie Kirsche,6 Sergey Aganezov,6 Michael C. Schatz,6 Giuseppe Narzisi,7 Marta Byrska-Bishop,7 Wayne Clarke,7 Uday S. Evani,7 Charles Markello,8 Kishwar Shafin,8 Xin Zhou,9 Arend Sidow,10 Vikas Bansal,11 Peter Ebert,12 Tobias Marschall,12 Peter Lansdorp,13 Vincent Hanlon,13 Carl-Adam Mattsson,13 Alvaro Martinez Barrio,14 Ian T Fiddes,14 Chunlin Xiao,15 Arkarachai Fungtammasan,16 Chen-Shan Chin,16 Aaron M Wenger,17 William J Rowell,17 Fritz J Sedlazeck,2 Andrew Carroll,18 Marc Salit,19,20 Justin M Zook1,20,21,*

Summary

Initial submission: Received : December 7th 2020
Scientific editor: Orli Bahcall and Judith Nicholson

First round of review: Number of reviewers: 3
Revision invited : June 22nd 2021
Revision received : September 30th 2021

Second round of review: Number of reviewers: 2
Accepted : March 2022

Data freely available: Yes

Code freely available: Yes

This transparent peer review record is not systematically proofread, type-set, or edited. Special characters, formatting, and equations may fail to render properly. Standard procedural text within the editor's letters has been deleted for the sake of brevity, but all official correspondence specific to the manuscript has been preserved.

Referees' reports, first round of review

Reviewer #1

A very nice upgrade of the GIAB truth sets, this is an extremely valuable contribution to the community.

Questions/Comments:

- How does the Mendelian inconsistent fraction compare to v3.3.2?
- In discussing Mendelian inconsistencies, you state that "We exclude these Mendelian inconsistencies that are unlikely to have a biological origin from the v4.2.1 benchmark regions of each member of the trio." It is not clear to me from language in the manuscript whether non-biological inconsistencies in category "(3) some overlapping complex variants in the MHC that were correctly called in the trio but the different representations were not reconciled by our method" are included or excluded.
- The manuscript is predominantly reporting v4.2.1, however in the "Evaluation of the benchmark" section it switches to discussing v4.1.
- It is mentioned that the number of FN for a standard short read call set increases by a factor of around 8.5 compared to v3.3.2. Does the number of FP increase similarly as well?
- One of the main advantages of the separate synthetic diploid benchmark mentioned in the introduction has been its ability to cover exactly these more difficult to call regions since it was produced using long reads. This has been observable in the past in the significantly higher FP and FN rates when benchmarking variant calling pipelines against syndip as compared to against GIAB v3.3.2, as well as in the larger fraction of the genome covered by the syndip high confidence region. It seems that this upgrade to GIAB likely negates that difference, and a direct comparison of FP/FN rates and genome coverage to demonstrate that would be interesting.
- I found Figure 3 a bit confusing. Based on the text, I was expecting this figure to visually demonstrate the increased PacBio and ONT coverage which is the signal of a duplication. However, the figure is zoomed in to only the duplicated region so that there is no normal coverage level to compare to. I would find a visual which

can demonstrate the increased coverage corresponding with the excluded regions more helpful.

- In the section on potential CNV detection, when describing the Elliptical Outlier Boundary method, an equation may be clearer than text.

- In the section on Benchmark evaluation, I would be interested to see how many of the evaluated FP/FNs are new to v4.2.1, as opposed to also being in v3.3.2.

Reviewer #2

GENERAL COMMENTS AND FEEDBACK

The work of GIAB to provide gold-standard variant call sets is of incredible value and the authors should be thanked for their work in this area.

Gold-standard call sets not only allow benchmarking of new pipelines but, when used as training sets for VQSR or other machine-learning based filtering approaches, have a massive impact and utility across almost all variant call sets where filtering is standard.

The utility of such data sets is added to by the fact that at present such gold-standard data sets are rare, making any additional or improved data useful.

This study looks at extending beyond the regions of the genome that have been previously assayable. While in terms of percentage of the genome covered it could be argued that these improvements are marginal, it is adding data in regions that could not previously be accessed and where the most widely used short read technologies struggle. This makes this data set of additional interest and adds to its value.

Looking critically at the manuscript, however, I do see scope for improvement in the areas that I am considering.

Sample choice and number:

I can see no rationale offered for the choice of samples used in this work.

The trio the authors have used (proband HG002) is openly consented and cell lines are available. However, the same is true for the trio including NA12878 as the proband. In addition, for NA12878, there are further samples from the same family (additional samples not openly consented but available) and the sample forms part of both a wider set of openly consented samples from the same population (CEU) and part of a much wider and heavily analysed collection (1000 Genomes). The consent for NA12878 (and indeed for all 1000 Genomes samples) was designed to enable exactly this sort of work and free sharing and reuse in any domain (academia, industry, etc.), as is publicly documented. In this context, it would have seemed preferable to overlap with a substantial and widely used reference collection, affording greater flexibility.

While the data presented is welcome, it would have been great to see this work extended across more samples. One could imagine, for example, a resource using the three trios presented in Chaisson et al (2019) and the NA12878 trio adding massively to this data set. In addition, some of the types of data used in this work already exist and are publicly available for these samples, so it would seem plausible to extend the work (although noting the additional labour for manual curation and Sanger validation). This would also have had the benefit of including at least one sample of African ancestry. I believe that, at present, GIAB provides no gold-standard call set for a genome of African ancestry, despite the increased levels of variation seen in such genomes. Again, such samples exist with cell lines and appropriate consent.

The limited nature of the samples included could also raise concerns for those wishing to use this as a truth set for machine-learning training purposes, where a broader data set would be imagined to be preferable.

Information on the sample selection process, describing the criteria used, would be beneficial to the manuscript.

Manuscript oversights:

I noticed the following issues:

1. Author affiliations include one numbered with 19 but it appears not to apply to any author.
2. In the first paragraph of the introduction, the following appears: "As these new technologies and methods access increasingly challenging regions of the genome, including many known medically relevant genes that were excluded from these previous benchmarks. " This doesn't read as a sentence. As X what? It seems incomplete
3. The methods note that data from Ensembl Biomart was used but no citation is included. Ensembl publish in the database issue of Nucleic Acids Research each year and should be cited appropriately.

I have not examined the whole manuscript closely for such issues but these gave the impression that the manuscript would benefit from greater attention.

Some aspects of the manuscript could be clearer. For example, VDJ is used as a section header but it was initially unclear to me what VDJ meant in the context (although I did jump ahead to the methods).

NOT REVIEWED

I do not have expertise in the analysis of long or linked read data, so am offering no comment on the analysis. As this represents a substantial portion of the manuscript, I would recommend seeking another review.

SUMMARY

GIAB gold-standard data sets are incredibly valuable. The use of long and linked reads to extend the regions of the genome covered is therefore welcome.

It would be nice to see this work extended across further samples, for the sample selection process to be addressed and the manuscript given some further attention.

Reviewer #3

This paper represents a significant improvement to the small variant calling benchmark data resource GiaB. It is important to be able to assess variant calling across as many genomic contexts as possible.

The section "More FNs are identified by the new benchmark" is really just setting up a straw man. Of course this is true. If you are going to include that section, you should at least give results for all variant calling / technology setups.

Figure 5 needs more clarification. How many FPs and FNs were manually curated in each set? After several rereads it makes sense. At least add totals to each.

As a point of minor clarification, in response to "v4.2.1 uses 10x Genomics linked-read data and the variant calls from the LongRanger pipeline6

,

in place of the conservative, haplotype-separated GATK calls from 10x Genomics used in v3.3.2" - longranger has a haplotype-separated step. It calls variants,

phases (and one phase state is "phasing inconsistent" and those variants are determined to either be homozygous ref or alt), separates reads and recalls variants. The discrepancies between those two vcfs are gone through and read data is aligned to each to determine which call is more accurate. Anyway, the minor clarification is that longranger also uses haplotype separation. But it is probably not relevant enough to mention it.

Figure 6 is also somewhat confusing. I would expect a color where things are included, not excluded.

Authors' response to the first round of review

Thank you for the helpful reviews. We have included our responses in blue below

Reviewers' Comments: Reviewer #1: A very nice upgrade of the GIAB truth sets, this is an extremely valuable contribution to the community.

Questions/Comments: - How does the Mendelian inconsistent fraction compare to v3.3.2?

- we added the fraction for v3.3.2 as well "This identified 2,502 Mendelian inconsistencies out of the 4,968,730 variants in at least one member of the trio and in the intersection of the benchmark regions for the trio (0.05%), similar to the rate for v3.3.2 (2,494 out of 4,383,371, or 0.05%)."

- In discussing Mendelian inconsistencies, you state that "We exclude these Mendelian inconsistencies that are unlikely to have a biological origin from the v4.2.1 benchmark regions of each member of the trio." It is not clear to me from language in the manuscript whether nonbiological inconsistencies in category "(3) some overlapping complex variants in the MHC that were correctly called in the trio but the different representations were not reconciled by our method" are included or excluded.

- We now clarify that these are excluded "We exclude all Mendelian inconsistencies that are not heterozygous in the son and homozygous reference in both parents from the v4.2.1 benchmark regions of each member of the trio, because most are unlikely to have a biological origin"

- The manuscript is predominantly reporting v4.2.1, however in the "Evaluation of the benchmark" section it switches to discussing v4.1.

- This manual evaluation was performed on v4.1, and only small improvements were made to make v4.2.1 for all samples, so we've added sentences about this: "We performed the community evaluation on v4.1 for HG002. Based on this evaluation, we made small improvements to Response to Reviewers generate v4.2.1 for HG002, as well as for the other 6 samples (Supplementary Note 2). v4.2.1 is the version described in the rest of this manuscript for all samples."

- It is mentioned that the number of FN for a standard short read call set increases by a factor of around 8.5 compared to v3.3.2. Does the number of FP increase similarly as well?

- The FP rate is a bit more nuanced and is an interesting point, so we've added the following text: "For a standard short read-based call set (Ill GATK-BWA in Figure 5), the number of SNVs missed (even when including filtered variants) was 8.5 times higher when benchmarking against v4.2.1 than against v3.3.2 (16,615 vs. 1,960). The difference is largely due to false negative SNVs in regions of low mappability and segmental duplications with 15,220 in v4.2.1 vs. 1,381 in v3.3.2. When counting conservatively filtered SNVs as false negatives, v4.2.1 detected 71,165 more errors (183,568 in v4.2.1 vs. 112,403 in v3.3.2), similar to the increases seen with the noisy longread-based syndip benchmark.(Li et al. 2018) Also similar to syndip, the number of false positive SNVs was higher for v4.2.1 (25,328) than v3.3.2 (13,788) before conservative filtering. However, the number of false positive SNVs was actually lower for v4.2.1 (1,539) than v3.3.2 (2,370) after conservative filtering, likely due to removal of potential structural and copy

number variants in v4.2.1. Relative to syndip, v4.2.1 for HG002 covers about 1 % fewer autosomal bases in GRCh38 but 16 % more bases in regions of low mappability and segmental duplications. The more challenging variants and regions included in v4.2.1 enables further optimization and development of variant callers in segmental duplications and low mappability regions.”

- One of the main advantages of the separate synthetic diploid benchmark mentioned in the introduction has been its ability to cover exactly these more difficult to call regions since it was produced using long reads. This has been observable in the past in the significantly higher FP and FN rates when benchmarking variant calling pipelines against syndip as compared to against GIAB v3.3.2, as well as in the larger fraction of the genome covered by the syndip high confidence region. It seems that this upgrade to GIAB likely negates that difference, and a direct comparison of FP/FN rates and genome coverage to demonstrate that would be interesting.

- While a direct comparison of FP/FNs rates was not possible due to differences in datasets and callsets available for the samples, we agree that the comparison to syndip is interesting and have added the results in the quote in our answer to the previous question. We also very recently developed a diploid assembly-based benchmark for challenging medically relevant genes that builds on the ideas in syndip, and the highly accurate long reads and assembly methods now available enabled even further characterization of challenging segmental duplications.

- I found Figure 3 a bit confusing. Based on the text, I was expecting this figure to visually demonstrate the increased PacBio and ONT coverage which is the signal of a duplication. However, the figure is zoomed in to only the duplicated region so that there is no normal coverage level to compare to. I would find a visual which can demonstrate the increased coverage corresponding with the excluded regions more helpful.

- We have now expanded Fig 3 to show the entire KIR regions as suggested, making the coverage variability clearer, and we cleaned up and simplified the tracks to make it more concise and easier to understand.

- In the section on potential CNV detection, when describing the Elliptical Outlier Boundary method, an equation may be clearer than text.

- Thank you for this suggestion. We have added this equation to the text: “We then found regions that had outlier coverage in PacBio HiFi and/or ONT. To do so, as described in the equations below, we (1) divided the PacBio HiFi coverage of each window by the median depth HiFi depth and squared it; (2) divided the ONT coverage of each window by the median depth ONT depth and squared it; (3) summed those values; and (4) took the square root of the sum. We found the third quartile and interquartile range for those transformed window coverage values. Finally, we found windows with coverage greater than the third quartile plus 1.5 times the IQR. In the equations below, WindowHiFiDepth, WindowONTDepth, and EllipticalValues are vectors, while MedianHiFiDepth, MedianONTDepth, and ThresholdEllipticalOutlier are scalars.”

- In the section on Benchmark evaluation, I would be interested to see how many of the evaluated FP/FNs are new to v4.2.1, as opposed to also being in v3.3.2.

- we have added the following statement “Overall, 433 of 452 (96%) curated FP and FN SNVs and Indels inside v3.3.2 benchmark regions and 422 of 463 (91%) outside v3.3.2 benchmark regions were determined to be correct in the v4.1 benchmark.”

Reviewer #2: GENERAL COMMENTS AND FEEDBACK The work of GIAB to provide gold-standard variant call sets is of incredible value and the authors should be thanked for their work in this area. Gold-standard call sets not only allow benchmarking of new pipelines but, when used as training sets for VQSR or other machinelearning based filtering approaches, have a massive impact and utility across almost all variant call sets where filtering is standard. The utility of such data sets is added to by the fact that at present such gold-standard data sets are rare, making any additional or improved data useful. This study looks at extending beyond the regions of the genome that have been previously assayable. While in terms of percentage of the genome covered it could be argued that these improvements are marginal, it is adding data in regions that could not previously be accessed and where the most widely used short read technologies struggle. This makes this data set of additional interest and adds to its value.

- thank you for your positive feedback. We agree that the increased inclusion of challenging regions is the primary and important value of this new benchmark.

Looking critically at the manuscript, however, I do see scope for improvement in the areas that I am considering.

Sample choice and number: I can see no rationale offered for the choice of samples used in this work. The trio the authors have used (proband HG002) is openly consented and cell lines are available. However, the same is true for the trio including NA12878 as the proband. In addition, for NA12878, there are further samples from the same family (additional samples not openly consented but available) and the sample forms part of both a wider set of openly consented samples from the same population (CEU) and part of a much wider and heavily analysed collection (1000 Genomes). The consent for NA12878 (and indeed for all 1000 Genomes samples) was designed to enable exactly this sort of work and free sharing and reuse in any domain (academia, industry, etc.), as is publicly documented. In this context, it would have seemed preferable to overlap with a substantial and widely used reference collection, affording greater flexibility.

- Thank you for highlighting this important question. While we acknowledge the advantage of working with 1000 Genomes samples, and in fact used NA12878 as our pilot sample, GIAB held extensive, public discussions about sample selection and chose the other 6 samples from the Personal Genome Project because they have a broader consent (e.g., companies cannot develop products that include NA12878 DNA, as they have for GIAB/PGP samples). We've clarified this under Materials Availability as "The Genome in a Bottle Consortium selected these seven genomes for characterization because the pilot HG001 had extensive pre-existing public data, and HG002 to HG007 are two trios from the Personal Genome Project that have a broader consent that permits commercial redistribution and recontacting participants for further sample collection."

While the data presented is welcome, it would have been great to see this work extended across more samples. One could imagine, for example, a resource using the three trios presented in Chaisson et al (2019) and the NA12878 trio adding massively to this data set. In addition, some of the types of data used in this work already exist and are publicly available for these samples, so it would seem plausible to extend the work (although noting the additional labour for manual curation and Sanger validation). This would also have had the benefit of including at least one sample of African ancestry. I believe that, at present, GIAB provides no gold-standard call set for a genome of African ancestry, despite the increased levels of variation seen in such genomes. Again, such samples exist with cell lines and appropriate consent.

- Similar to what the reviewer suggests, we have been working towards similar v4.2.1 benchmarks for the other 4 GIAB samples: NA12878 and a Han Chinese trio from the Personal Genome Project. We have now released these and include them in the revised manuscript. We do not yet include any samples of African ancestry, but we are coordinating with the Human Pangenome Reference Consortium to include more diversity in future benchmarks, which we agree will be very valuable.

The limited nature of the samples included could also raise concerns for those wishing to use this as a truth set for machine-learning training purposes, where a broader data set would be imagined to be preferable.

- We have had discussions about this on the public GIAB machine learning calls, and this was one of the motivations for developing benchmarks for the remaining GIAB samples. We hope to explore the impact of ancestry on variant calling in the near future.

Information on the sample selection process, describing the criteria used, would be beneficial to the manuscript.

- As noted above, we have added this under Materials Availability

Manuscript oversights: I noticed the following issues: 1. Author affiliations include one numbered with 19 but it appears not to apply to any author.

- this has been corrected

2. In the first paragraph of the introduction, the following appears: "As these new technologies and methods access increasingly challenging regions of the genome, including many known medically relevant genes that were excluded from these previous benchmarks." This doesn't read as a sentence. As X what? It seems incomplete

- Thank you for pointing this out. This has been corrected to "As these new technologies and methods accessed increasingly challenging regions of the genome, studies highlighted many known medically

relevant genes that were excluded from these previous benchmarks”

3. The methods note that data from Ensembl Biomart was used but no citation is included. Ensembl publish in the database issue of Nucleic Acids Research each year and should be cited appropriately.

- We have now added the citation to NAR.

I have not examined the whole manuscript closely for such issues but these gave the impression that the manuscript would benefit from greater attention.

- We have read through the manuscript again and corrected any other outstanding issues

Some aspects of the manuscript could be clearer. For example, VDJ is used as a section header but it was initially unclear to me what VDJ meant in the context (although I did jump ahead to the methods).

- We have updated the formatting of sub-headings to make it clearer VDJ and others fall under “Regions excluded from the benchmark”, and also added more context to the heading: “VDJ region subject to somatic recombination”

NOT REVIEWED I do not have expertise in the analysis of long or linked read data, so am offering no comment on the analysis. As this represents a substantial portion of the manuscript, I would recommend seeking another review.

SUMMARY

GIAB gold-standard data sets are incredibly valuable. The use of long and linked reads to extend the regions of the genome covered is therefore welcome. It would be nice to see this work extended across further samples, for the sample selection process to be addressed and the manuscript given some further attention.

- Thank you for the helpful review, and we agree that the work we have done to extend to the 4 additional samples will be valuable, and we plan to expand to samples of other ancestries using new assembly-based approaches.

Reviewer #3: This paper represents a significant improvement to the small variant calling benchmark data resource GiaB. It is important to be able to assess variant calling across as many genomic contexts as possible. The section “More FNs are identified by the new benchmark” is really just setting up a straw man. Of course this is true. If you are going to include that section, you should at least give results for all variant calling / technology setups.

- We recognize now that this section heading understates the claim we are making. We have changed the heading to “New benchmark regions are enriched for false negatives” to clarify that these results help demonstrate that the new benchmark regions really are more “difficult” in practice in that they contain more variant calling errors. We had originally done this analysis across methods but GIAB consortium members felt this came across as a comparison of methods, which are rapidly evolving. We have substantially revised this section based on all the reviewers’ feedback and have also added a reference to the companion paper about the precisionFDA V2 challenge: “The SNV error rates of the Truth Challenge V1 winners decrease by as much as 10-fold when benchmarked against the new V4.2 benchmark set, compared to the V3.2 benchmark set used to evaluate the first truth challenge”

Figure 5 needs more clarification. How many FPs and FNs were manually curated in each set? After several rereads it makes sense. At least add totals to each.

- We’ve now clarified this in the caption: “(A) For each callset, we curated 20 FPs and 20 FNs, and this shows the proportion of curated FP and FN variants where the benchmark set was correct and the query callset was incorrect. The dashed black line indicates the desired majority threshold, 50%. Half of the curated variants were from GRCh37 and half were from GRCh38.”

As a point of minor clarification, in response to “v4.2.1 uses 10x Genomics linked-read data and the variant calls from the LongRanger pipeline in place of the conservative, haplotype-separated GATK calls from 10x Genomics used in v3.3.2” - longranger has a haplotype-separated step. It calls variants, phases (and one phase state is “phasing inconsistent” and those variants are determined to either be homozygous ref or alt), separates reads and recalls variants. The discrepancies between those two vcfs are gone through and read data is aligned to each to determine which call is more accurate. Anyway, the minor clarification is that longranger also uses haplotype separation. But it is probably not relevant enough to mention it.

- Thanks for pointing out this confusing language. As the reviewer suggests, we didn’t mean to say that

Longranger doesn't haplotype-separate the reads (though it did not do this yet at the time of making v3.3.2, which is why we did our own haplotype separation). We have clarified this in "v4.2.1 uses 10x Genomics linked-read data and the variant calls from the LongRanger pipeline6, which makes calls both with and without using information from partitioning reads into haplotypes. In v3.3.2, we used conservative, haplotype-separated GATK calls from 10x Genomics, where calls were only made separately on each haplotype and coverage from both haplotypes was required "

Figure 6 is also somewhat confusing. I would expect a color where things are included, not excluded. Was simplified and now included regions are indicated by black rectangles.

Referees' report, second round of review

Reviewer #1

Appreciate the revisions to the original, and good to see more samples being updated. Thank you for the work to make such a valuable resource available to the community.

A few minor comments:

- In the intro, you write "Our new benchmark includes more than ... 16% new exonic variants..."

It isn't completely clear to me what this means. Has the new benchmark increased the number of exonic variants by 16% (this is my guess), or are 16% of new variants exonic (I doubt it, but the wording isn't really clear).

- In Fig 5, the labels "yes", "no", "partial", and "unsure" aren't clearly defined. I think "yes", for example, means that the variant was deemed to be correct by manual curation, but I think the labelling could be clearer.

-In the section "New benchmark regions enriched for false negatives", the "conservatively filtered" calls seem EXTREMELY conservatively filtered (100x more FN than FP!). As such, it's not clear to me how much the conclusions drawn based on the filtered data illustrate features of the truth set as opposed to the behavior of the filtering, especially since the unfiltered calls have the opposite FP behavior

(more FP in v4.2.1 than v3.3.2). It might be more convincing to see numbers from multiple different technologies/pipelines, or perhaps roc curves from a single pipeline.

Reviewer #3

Thank you for this valuable data resource for the community. The clarifications and other edits are very helpful.

Authors' response to the second round of review

Reviewer #1:

Appreciate the revisions to the original, and good to see more samples being updated. Thank you for the work to make such a valuable resource available to the community.

Thank you for the helpful review, and we now have added limitations of samples to the "Limitations" section after the Discussion: " Furthermore, the genomes characterized in this work come from individuals of European, Ashkenazi Jewish, and Han Chinese ancestry, and future benchmarks are needed to understand any potential differences in variant benchmarking for other ancestries. "

A few minor comments: - In the intro, you write "Our new benchmark includes more than ... 16% new exonic variants..." It isn't completely clear to me what this means. Has the new benchmark increased the number of exonic variants by 16% (this is my guess), or are 16% of new variants exonic (I doubt it, but the wording isn't really clear).

You are correct that it is the first case. We've revised this to "These benchmarks add more than 300,000 SNVs and 50,000 indels, and include 16% more exonic variants, many in challenging, clinically relevant genes not previously covered"

- In Fig 5, the labels "yes", "no", "partial", and "unsure" aren't clearly defined. I think "yes", for example, means that the variant was deemed to be correct by manual curation, but I think the labelling could be clearer.

We have clarified the legend to say " (B) Breakdown of the total number of variants by category determined during manual curation, where Benchmark

Curation bar indicates whether the benchmark variant and genotype were determined to be correct, and Query Curations color indicates whether the query variant and genotype were determined to be correct. Panel B excludes variants from panel A where the benchmark was deemed correct and query incorrect, and shows most of these sites were difficult to curate. ”

-In the section "New benchmark regions enriched for false negatives", the "conservatively filtered" calls seem EXTREMELY conservatively filtered (100x more FN than FP!). As such, it's not clear to me how much the conclusions drawn based on the filtered data illustrate features of the truth set as opposed to the behavior of the filtering, especially since the unfiltered calls have the opposite FP behavior (more FP in v4.2.1 than v3.3.2). It might be more convincing to see numbers from multiple different technologies/pipelines, or perhaps roc curves from a single pipeline.

We agree with the reviewer that these results may differ in magnitude based on exactly how variants are filtered. Our companion manuscript about the precisionFDA challenge discusses these results for more variant calling pipelines, so we've added more about these and referenced the companion manuscript in this text: "Comparison of the results from the first and second precisionFDA challenges (based on v3.2 and v4.2, respectively), demonstrated similar changes in performance when expanding the benchmark; the combined false positive and false negative rates for SNVs increased by 2-fold to 10-fold when the five top performers of the first challenge were benchmarked against v4.2.